# Global burden and regional disparities of rheumatoid arthritis among the working-age population: A comprehensive analysis from 1990 to 2021 with projections to 2040

Jun Li ⓘ*, Zhiyong Li, Chengluo Hao, Xiangrui Chen

The Third People's Hospital of Zigong, Zigong, Sichuan, China

* jijisister163@163.com

## Abstract

### Objective

To evaluate the age-standardized incidence (ASIR), prevalence (ASPR), death (ASDR), and disability-adjusted life year (DALY) rates of rheumatoid arthritis (RA) among the working-age population from 1990 to 2021.

### Methods

The data is sourced from the Global Burden of Disease 2021. Estimated annual percentage change (EAPC) was utilized to assess temporal trends. Decomposition analysis was conducted to identify the driving factors underlying burden changes. The Slope Index of Inequality and the Concentration Index were employed to evaluate cross-country inequalities.

### Results

In 2021, there were 11.88 million cases of RA in the working-age population globally and an ASPR of 222.68 per 100,000 population. The ASIR was 14.09 per 100,000 population (95% uncertainty interval [UI]: 8.97−20.19), while the ASDR was 0.13 per 100,000 population (95% UI: 0.11−0.15), with an EAPC of −1.59 (95% confidence interval [CI]: −1.73 to −1.45), indicating a sustained decline in RA ASDR. The age-standardized DALYs rate was 34.54 per 100,000 population (95% UI: 23.90−48.67), with an EAPC of 0.15 (95% CI: 0.10 to 0.20). Regionally, high Socio-Demographic Index (SDI) regions exhibited the highest ASPR, ASIR, and age-standardized DALYs rates, suggesting a greater overall burden of RA. Interestingly, middle SDI regions showed the highest ASDR, potentially indicating differences in disease management and access to care that impact mortality despite a lower overall burden compared to high SDI regions. Decomposition analysis identified

**Data availability statement:** Data is not publicly available and can only be accessed after an account has been registered. All datasets are available from the GBD database (https://vizhub. healthdata.org/gbd-results/).

**Funding:** The author(s) received no specific funding for this work.

**Competing interests:** The authors have declared that no competing interests exist.

population growth as the primary driver of the increasing RA burden. Cross-national inequality analysis revealed that RA burden remains concentrated in high SDI countries, though overall health inequality has declined.

## Conclusions

The substantial global burden and regional disparities of RA in the working-age population necessitate targeted interventions. High SDI regions require strategies focusing on early diagnosis and optimal management to reduce the high burden. Elevated mortality in middle SDI regions demands improved access to effective treatment. These findings underscore the need for SDI-tailored public health approaches to address the specific challenges in each context.

---

## Introduction

Rheumatoid arthritis (RA) is a prevalent autoimmune disorder causing chronic, symmetrical joint inflammation, leading to cartilage and bone damage and impaired joint function [1]. Beyond joints, RA can cause systemic complications like cardiovascular, pulmonary, and mental health disorders, significantly impacting quality of life [2]. In 2020, approximately 17.6 million individuals worldwide were diagnosed with RA, with a prevalence of 208.8 per 100,000 population and a mortality rate of 0.47 per 100,000 population, exhibiting notable regional variations [3]. In low- and middle-income countries, the early diagnosis and treatment of RA still face significant challenges [4]. The high prevalence of RA among working-age populations is particularly concerning, as research identifies it as a leading cause of reduced work capacity and early retirement, imposing a significant economic burden on patients and their families [5,6]. From a global perspective, RA's adverse impact on economic productivity is increasingly apparent, as patients face reduced full-time employment opportunities, rising healthcare costs, and immeasurable societal losses [7–9].

Current research predominantly focuses on the etiology and advancements in biological therapies for RA. Significant progress has been made in understanding how various treatment approaches, such as biologics, alleviate symptoms and modify disease progression [10]. However, these studies are largely concentrated in high-income countries, leaving the realities of RA in low- and middle-income populations underrepresented. Notably, the burden of RA on working-age populations and its associated economic consequences remains insufficiently explored on a global scale. Existing research on productivity loss due to RA is often restricted to specific regions or limited sample sizes, lacking a comprehensive and multidimensional perspective [11]. Furthermore, economic evaluations of RA management strategies, including cost-effectiveness analyses and resource allocation studies, are relatively scarce. These gaps in the literature impede the development of effective public health policies and highlight the urgent need for a comprehensive global analysis. Therefore, this study utilizes the Global Burden of Disease 2021 (GBD 2021) data to address these limitations by analyzing the global RA burden among the working-age population and the associated regional disparities.

This study aims to (1) quantify the impact of RA on the labor force and evaluate trends in disease burden over a 32-year period, (2) examine disparities in disease burden across regions and income levels, and (3) provide precise data to inform policymakers, facilitating the optimization of resource allocation. The unique contribution of this research lies in offering a novel perspective on the effects of RA in working-age populations for the first time. Moreover, the findings will serve as a foundation for developing targeted intervention strategies, promoting comprehensive health management, and social integration for individuals living with RA.

## Methods

### Data source

The GBD database is the most extensive and comprehensive resource for estimating disease burden worldwide. The latest iteration, GBD 2021, encompasses 23 age groups, ranging from neonates to individuals aged 95 years and older, and evaluates the burden of 371 diseases and injuries, as well as 88 risk factors, across 204 countries and territories from 1990 to 2021 [12]. GBD 2021 employs the Bayesian meta-regression tool DisMod-MR 2.1 to estimate RA incidence, prevalence, mortality, and disability-adjusted life year (DALY) by year, age, and sex for 204 countries and territories [3]. Data are reported by country, region, and super-region, categorized based on epidemiological similarity and geographic proximity. For most disease models in GBD, data are unavailable for certain countries or regions. In such cases, burden estimates are derived using available data from super-regional priors. DisMod-MR 2.1 facilitates this process through hierarchical geographic cascade modeling. Initial models are constructed at the global, super-regional, or regional level for areas with available data. These results then serve as Bayesian priors for subsequent models at lower levels. For regions without direct data, estimates are informed by priors from higher-level regions, ensuring consistency and accuracy throughout the modeling process [3].

The dataset utilized in this research was accessed through the Global Health Data Exchange query tool, available at https://vizhub.healthdata.org/gbd-results/. As we utilized the final, processed estimates provided within the GBD 2021 results tool, no further adjustments or handling of missing data were performed by the authors. The institutional review board of the Third People's Hospital of Zigong determined that the study did not need ethics approval because it used publicly available data.

### Rheumatoid arthritis definition

RA is a chronic autoimmune disease typically diagnosed using the 1987 classification criteria established by the American College of Rheumatology [13]. According to these criteria, a diagnosis of RA requires the presence of at least four out of seven clinical manifestations: morning stiffness, arthritis in three or more joint areas, symmetric arthritis, arthritis in hand joints, rheumatoid nodules, positive serum rheumatoid factor, and radiographic evidence of joint damage. Notably, the first four criteria must persist for a minimum of six weeks to meet the diagnostic threshold. In the GBD 2021 study, RA cases are identified using the International Classification of Diseases (ICD) coding system. Specifically, ICD-10 codes M05-M06.9 and M08.0-M08.8, along with the corresponding ICD-9-CM codes 714–714.3 and 714.8–714.9, are applied to analyze medical claims data and classify RA cases consistently [3].

### Study population

The working-age population, typically defined as individuals aged 15–64, plays a critical role in socioeconomic systems as the primary participants in labor and employment activities. The health of this group directly impacts labor force participation rates and productivity, thereby influencing overall economic development. We categorize the population into ten distinct age groups—15–19, 20–24, 25–29, 30–34, 35–39, 40–44, 45–49, 50–54, 55–59, and 60–64 years—to analyze the age-specific burden of RA within this demographic. Data on incidence, prevalence, mortality, and DALYs are collected and analyzed across these age groups, with additional stratifications based on geographic regions and Socio-Demographic

Index (SDI). This approach provides a comprehensive understanding of the burden of RA among the working-age population in varying contexts.

## Socio-demographic index

The SDI is a composite measure that reflects a country's income per capita, average educational attainment, and total fertility rate under the age of 25. The higher the SDI value, the better the socio-economic development [3]. For this analysis, the country-level SDI estimates were used to determine thresholds for dividing the 204 countries and territories included in the GBD 2021 dataset into five SDI quintiles: low, low-middle, middle, high-middle, and high. This stratification allows for a detailed examination of health outcomes and disease burden across varying levels of socioeconomic development.

## Statistical analysis

To comprehensively assess the burden of RA among the working-age population, a descriptive analysis was conducted at global, regional, and national levels. The analysis visually illustrates trends in the number of RA cases, age-standardized rates of incidence (ASIR), prevalence (ASPR), death (ASDR), and DALYs from 1990 to 2021. Subgroup analyses were further performed by sex and age groups, both globally and across the five SDI quintiles. Each rate is expressed per 100,000 population and reported with 95% uncertainty intervals (UIs) calculated using the GBD algorithm.

Exploring temporal trends in disease patterns is an indispensable component of epidemiology, providing critical insights for the development of precise prevention strategies. To quantify global trends in the burden of RA among the working-age population, estimated annual percentage changes (EAPCs) derived from linear regression models were employed. EAPCs serve as a robust metric for monitoring changes in ASRs, making them particularly suitable for tracking shifts in disease patterns [14]. The EAPCs are presented alongside their 95% confidence intervals (CIs). Trends were determined by evaluating the EAPC and its corresponding 95% CI: a positive EAPC with a 95% CI above zero indicated an upward trend, while a negative EAPC with a 95% CI below zero signified a downward trend. Additionally, a smoothed curve-fitting model was applied to describe the relationship between the burden of RA and the SDI across different countries and regions [15].

We also examined trends in the burden of RA associated with age, period, and cohort effects. The potential two-way interactions among age, period, and cohort were initially explored to evaluate their influence on ASRs. Given the complex interactions among these temporal factors, it is challenging to isolate their relative contributions to the observed trends in ASRs. To address this complexity, an age-period-cohort (APC) model was employed to simultaneously investigate temporal changes in the risks of incidence, prevalence, mortality, and DALY rates across three time dimensions. Due to the inherent multicollinearity among age, period, and cohort, a refined APC model incorporating principal component regression analysis was utilized. This approach allowed us to disentangle the distinct effects of these three temporal trends and provide more reliable estimates [16,17]. The model yielded estimated coefficients for age, period, and cohort effects, which were subsequently exponentiated to calculate relative risks (RRs). These RRs represent the risk of ASIR, ASPR, ASDR, and age-standardized DALYs rate for a specific age, period, or cohort relative to the average comprehensive level across all ages, periods, or cohorts [18].

To understand the extent to which population growth, aging, and epidemiological changes have driven variations in the incidence, prevalence, deaths, and DALYs of RA among the working-age population over the past 32 years, a decomposition analysis was conducted by gender and region [18]. The contribution of each driving factor to changes in incidence, prevalence, deaths, and DALYs was defined as the impact of altering one factor while holding the other factors constant. In addition, monitoring cross-national inequalities can inform the development of tailored policies to address specific disease burdens in different regions. To achieve this, our study employed the Slope Index of Inequality (absolute inequality) and the Concentration Index (relative inequality) to evaluate disparities in the burden of RA among the working-age population across countries [18]. Finally, to predict changes in RA incidence, prevalence, mortality, and DALYs rates and

cases from 2022 to 2040 by sex, we utilized a Bayesian age-period-cohort (BAPC) model based on integrated nested Laplace approximation (INLA). This approach was chosen for its superior coverage and accuracy, which help to avoid the mixing and convergence issues commonly associated with traditional models [19–21]. All analyses and visualizations were conducted using JD_GBDR (V2.34.2, Jingding Medical Technology Co., Ltd.) and R statistical computing software (version 4.4.2, Vienna, Austria).

## Results

### Global burden

Detailed statistics are summarized in **Table 1**, **Fig 1**, and S1 Fig. In 2021, the global working-age population experienced a 90.52% increase in RA incidence since 1990, reaching 739,962.95 cases (95% UI: 471,404.72−1,060,372.08). The ASIR rose from 12.75 to 14.09 per 100,000 population (EAPC: 0.40, 95% CI: 0.37−0.44), indicating a consistent rise. In 2021, the global burden of RA in the working-age population reached 11,878,843.81 cases (95% UI: 9,521,211.43−14,623,475.28), an astonishing 108.39% increase from 1990. The ASPR rose from 195.05 per 100,000 population (95% UI: 154.69−242.77) in 1990 to 222.68 per 100,000 population (95% UI: 178.01−274.73) in 2021, with an EAPC of 0.53 (95% CI: 0.49 to 0.57). RA-related deaths in 2021 were 6,873.72 (95% UI: 5,830.26−7,956.85), with an ASDR of 0.13 per 100,000 population (EAPC: −1.59, 95% CI: −1.73 to −1.45), reflecting a sustained decline. Meanwhile, the age-standardized DALYs rate was 34.54 per 100,000 population (EAPC: 0.15, 95% CI: 0.10−0.20), showing a modest increase.

### Regional burden

The global burden of RA among the working-age population exhibited significant regional disparities, with a positive correlation to SDI levels (S2 Fig). The highest ASIR was observed in regions with high-middle SDI, at 19.99 per 100,000 population (95% UI: 13.34−27.76), whereas the lowest ASIR was reported in low SDI regions, at 7.91 per 100,000 population (95% UI: 4.89−11.63) (**Table 1**, **Fig 1**). However, regions with high SDI demonstrated the slowest growth, with an EAPC of 0.28 (95% CI: 0.19 to 0.36). In contrast, low-middle SDI regions exhibited the highest growth, with an EAPC of 0.99 (95% CI: 0.91 to 1.07) (**Table 1**, **Fig 1**). The ASIR findings were further supported by patterns in the ASPR, ASDR, and age-standardized DALYs rate, which emphasized regional differences. High SDI regions reported the highest ASPR and age-standardized DALYs rate, while middle SDI regions showed the highest ASDR. Specifically, high SDI regions recorded an ASPR of 293.62 per 100,000 population (95% UI: 240.22−354.79) and age-standardized DALYs rate of 43.01 per 100,000 population (95% UI: 29.66−60.66). In middle SDI regions, the ASDR reached 0.15 per 100,000 population (95% UI: 0.12−0.18) (**Table 1**, **Fig 1**). These findings collectively underscore the intricate relationship between socio-demographic factors and RA outcomes. While high SDI regions report the highest ASIR, ASPR, and age-standardized DALYs rate, they have made significant progress in reducing RA ASDR. Conversely, low-middle SDI regions exhibit the fastest-growing ASPR (EAPC: 1.00), indicating emerging challenges.

The ASIR of RA in the labor force of Andean Latin America reached 28.48 per 100,000 population (95% UI: 17.89−41.15), followed closely by Central Latin America at 26.61 per 100,000 population (95% UI: 17.26−37.18). Australasia also exhibited a high ASIR of 26.00 per 100,000 population (95% UI: 15.99−38.15) (**Table 1**). Similarly, these three regions continue to occupy the top ranks in ASPR for RA, with rates of 538.13 (95% UI: 438.86−648.63), 426.05 (95% UI: 347.11−512.88), and 335.82 (95% UI: 255.24−427.43), respectively (**Table 1**). In contrast, Oceania exhibited the lowest ASPR at 65.30 per 100,000 population (95% UI: 49.41−83.76) (**Table 1**). Moreover, Asia presents contrasting trends, with Central Asia showing the sharpest ASPR increase (EAPC: 1.38; 95% CI: 1.29 to 1.47), while high-income Asia Pacific has the slowest growth (EAPC: 0.08; 95% CI: −0.03 to 0.19) (**Table 1**). These starkly contrasting trends underscore the complexity and variability of RA prevalence in the labor force across different regions of Asia, highlighting the need for targeted interventions and further research to address these disparities.

**Table 1. Global and regional trends in the burden of rheumatoid arthritis among working-age population: incidence, prevalence, deaths, and DALYs (1990-2021).**

| Location | 1990 | | 2021 | | EAPC (95% CI) |
|---|---|---|---|---|---|
| | Number | ASR | Number | ASR | |
| **Incidence** | | | | | |
| Global | 383385.11 (243363.73, 564793.52) | 12.75 (8.01, 18.54) | 739962.95 (471404.72, 1060372.08) | 14.09 (8.97, 20.19) | 0.40 (0.37 to 0.44) |
| High SDI | 116290.00 (75875.59, 163294.88) | 19.07 (12.44, 26.79) | 169029.95 (113360.63, 234374.78) | 19.99 (13.34, 27.76) | 0.28 (0.19 to 0.36) |
| High-middle SDI | 85746.08 (52646.01, 125819.18) | 12.39 (7.61, 18.18) | 145729.65 (91763.57, 209935.18) | 14.92 (9.38, 21.48) | 0.66 (0.63 to 0.69) |
| Middle SDI | 120379.98 (73492.31, 177961.10) | 12.22 (7.42, 18.13) | 248877.79 (154887.14, 362600.67) | 14.53 (9.05, 21.15) | 0.62 (0.59 to 0.65) |
| Low-middle SDI | 51269.42 (31413.33, 75957.83) | 8.88 (5.44, 13.17) | 134719.23 (84514.30, 196407.46) | 11.66 (7.33, 17.00) | 0.99 (0.91 to 1.07) |
| Low SDI | 14398.20 (8660.63, 21591.11) | 6.51 (3.94, 9.72) | 41162.65 (25233.26, 60870.72) | 7.91 (4.89, 11.63) | 0.73 (0.62 to 0.84) |
| Andean Latin America | 3670.57 (2344.06, 5247.49) | 17.03 (10.71, 24.60) | 12257.74 (7719.41, 17694.99) | 28.48 (17.89, 41.15) | 1.80 (1.73 to 1.86) |
| Australasia | 3240.33 (2057.96, 4717.90) | 24.12 (15.35, 35.09) | 6092.17 (3791.86, 8873.35) | 26.00 (15.99, 38.15) | 0.35 (0.18 to 0.52) |
| Caribbean | 1563.21 (927.75, 2364.16) | 7.61 (4.49, 11.51) | 3019.66 (1811.79, 4502.59) | 9.56 (5.73, 14.22) | 0.73 (0.64 to 0.82) |
| Central Asia | 3655.03 (2246.53, 5326.73) | 9.09 (5.55, 13.23) | 8362.43 (5389.96, 11856.34) | 13.19 (8.48, 18.73) | 1.34 (1.24 to 1.45) |
| Central Europe | 11001.56 (6909.99, 15693.44) | 12.89 (8.10, 18.38) | 13110.72 (8379.23, 18537.99) | 15.38 (9.74, 21.88) | 0.66 (0.61 to 0.71) |
| Central Latin America | 17397.86 (10891.74, 24828.81) | 21.23 (13.25, 30.34) | 44907.42 (29119.78, 62725.83) | 26.61 (17.26, 37.18) | 0.68 (0.58 to 0.78) |
| Central Sub-Saharan Africa | 1600.11 (964.93, 2363.73) | 6.84 (4.21, 9.96) | 5243.95 (3243.60, 7629.81) | 8.37 (5.26, 12.04) | 0.69 (0.58 to 0.80) |
| East Asia | 112329.74 (66872.69, 168621.13) | 14.78 (8.80, 22.24) | 188848.63 (116495.34, 277140.77) | 16.87 (10.38, 24.72) | 0.52 (0.48 to 0.56) |
| Eastern Europe | 15713.17 (9707.73, 22758.98) | 10.30 (6.39, 14.88) | 17841.95 (11235.86, 25434.40) | 12.46 (7.85, 17.78) | 0.66 (0.63 to 0.69) |
| Eastern Sub-Saharan Africa | 4607.43 (2732.08, 6976.52) | 6.02 (3.64, 8.99) | 12876.46 (7736.68, 19166.47) | 6.73 (4.12, 9.89) | 0.40 (0.33 to 0.46) |
| High-income Asia Pacific | 28718.44 (17931.35, 41992.46) | 22.24 (13.86, 32.58) | 29496.34 (18752.83, 42084.07) | 20.20 (12.71, 29.00) | −0.12 (−0.24 to 0.00) |
| High-income North America | 33229.58 (22809.61, 44839.28) | 17.86 (12.27, 24.10) | 60436.02 (42334.64, 80939.88) | 21.47 (15.08, 28.64) | 0.77 (0.69 to 0.85) |
| North Africa and Middle East | 9610.41 (5910.25, 14112.83) | 5.25 (3.19, 7.75) | 31622.76 (19538.78, 45841.42) | 7.77 (4.80, 11.28) | 1.35 (1.30 to 1.41) |
| Oceania | 103.71 (57.94, 161.02) | 2.91 (1.62, 4.52) | 281.59 (161.72, 429.39) | 3.42 (1.96, 5.21) | 0.45 (0.39 to 0.51) |
| South Asia | 58252.60 (35266.46, 87418.17) | 10.60 (6.42, 15.92) | 166608.82 (102543.56, 246644.78) | 14.56 (8.98, 21.55) | 1.15 (1.03 to 1.27) |
| Southeast Asia | 10029.90 (5670.73, 15521.49) | 3.98 (2.26, 6.15) | 25621.89 (14842.66, 39104.05) | 5.30 (3.06, 8.09) | 1.00 (0.96 to 1.03) |
| Southern Latin America | 3463.80 (2090.41, 5106.26) | 11.57 (6.99, 17.03) | 8929.65 (5483.31, 12910.39) | 19.17 (11.75, 27.75) | 1.57 (1.47 to 1.67) |
| Southern Sub-Saharan Africa | 5920.73 (3948.85, 8220.22) | 19.92 (13.06, 28.00) | 9439.57 (6176.07, 13286.25) | 17.88 (11.60, 25.35) | −0.21 (−0.31 to −0.12) |
| Tropical Latin America | 10127.17 (6201.19, 15085.75) | 10.63 (6.41, 15.99) | 16193.23 (9934.35, 24016.72) | 10.58 (6.53, 15.63) | 0.19 (0.12 to 0.27) |
| Western Europe | 50921.64 (33068.75, 73038.80) | 18.36 (11.86, 26.41) | 68150.26 (45027.31, 96863.47) | 20.24 (13.21, 28.99) | 0.37 (0.30 to 0.44) |
| Western Sub-Saharan Africa | 3228.11 (1803.61, 5009.22) | 3.62 (2.02, 5.60) | 10621.68 (6113.76, 16133.31) | 4.51 (2.60, 6.85) | 0.78 (0.64 to 0.92) |
| **Prevalence** | | | | | |
| Global | 5700332.89 (4490981.36, 7135149.16) | 195.05 (154.69, 242.77) | 11878843.81 (9521211.43, 14623475.28) | 222.68 (178.01, 274.73) | 0.53 (0.49 to 0.57) |
| High SDI | 1649299.00 (1332519.91, 2021332.12) | 268.29 (216.37, 329.12) | 2590399.11 (2138302.78, 3105063.06) | 293.62 (240.22, 354.79) | 0.44 (0.36 to 0.52) |
| High-middle SDI | 1359598.83 (1068593.86, 1696169.78) | 199.88 (157.32, 248.87) | 2605744.97 (2098945.08, 3182446.96) | 245.22 (195.16, 302.60) | 0.72 (0.69 to 0.75) |
| Middle SDI | 1780085.86 (1375224.91, 2251488.52) | 198.03 (154.90, 247.83) | 4180912.57 (3325195.38, 5167274.69) | 235.70 (186.63, 292.39) | 0.63 (0.61 to 0.66) |
| Low-middle SDI | 710780.98 (542335.65, 911755.98) | 130.99 (101.32, 166.37) | 1940397.84 (1515553.38, 2452543.22) | 172.12 (135.43, 216.36) | 1.00 (0.91 to 1.10) |

Table 1. (Continued)

| Location | 1990 | | 2021 | | EAPC (95% CI) |
|---|---|---|---|---|---|
| | Number | ASR | Number | ASR | |
| Low SDI | 195699.01 (146183.31, 254794.05) | 95.10 (72.40, 121.95) | 553662.48 (417673.33, 715399.20) | 114.46 (88.24, 145.40) | 0.70 (0.58 to 0.81) |
| Andean Latin America | 60647.85 (48772.93, 73813.49) | 329.57 (268.80, 396.58) | 225072.72 (182885.44, 272171.94) | 538.13 (438.86, 648.63) | 1.74 (1.66 to 1.82) |
| Australasia | 38497.36 (29282.27, 49146.72) | 286.05 (217.63, 364.94) | 80433.35 (61921.71, 101315.32) | 335.82 (255.24, 427.43) | 0.60 (0.46 to 0.75) |
| Caribbean | 25177.35 (19274.24, 32021.83) | 134.22 (104.09, 168.95) | 56832.31 (44789.25, 70211.80) | 174.57 (137.00, 216.42) | 0.86 (0.77 to 0.95) |
| Central Asia | 59731.90 (46962.84, 73853.85) | 164.53 (130.90, 201.08) | 153644.35 (125353.39, 185280.87) | 240.81 (196.38, 290.55) | 1.38 (1.29 to 1.47) |
| Central Europe | 183191.72 (146217.18, 224126.31) | 203.68 (161.11, 250.85) | 229402.73 (184610.26, 279578.21) | 243.03 (192.28, 300.65) | 0.67 (0.62 to 0.73) |
| Central Latin America | 236287.81 (184617.68, 293374.89) | 329.82 (262.46, 403.03) | 720850.02 (587359.18, 867480.98) | 426.05 (347.11, 512.88) | 0.77 (0.65 to 0.88) |
| Central Sub-Saharan Africa | 21472.00 (16174.53, 27702.14) | 100.57 (77.58, 127.05) | 68465.23 (52263.92, 87105.86) | 120.20 (94.13, 149.83) | 0.60 (0.50 to 0.70) |
| East Asia | 1640976.26 (1254400.34, 2097332.14) | 230.92 (178.34, 292.57) | 3301940.12 (2622688.42, 4083356.00) | 269.45 (211.14, 337.08) | 0.59 (0.55 to 0.64) |
| Eastern Europe | 311305.01 (251650.79, 379059.83) | 187.62 (150.09, 230.22) | 387012.72 (318601.65, 463320.53) | 229.40 (185.52, 278.94) | 0.71 (0.67 to 0.76) |
| Eastern Sub-Saharan Africa | 59564.47 (43917.82, 78469.77) | 85.73 (65.05, 110.28) | 165694.40 (122570.55, 217458.90) | 96.26 (73.55, 122.98) | 0.41 (0.35 to 0.48) |
| High-income Asia Pacific | 369584.75 (284345.66, 470394.23) | 282.83 (217.08, 360.90) | 431194.60 (339396.86, 536252.67) | 276.81 (215.03, 348.68) | 0.08(−0.03 to 0.19) |
| High-income North America | 479727.93 (406670.14, 561384.20) | 259.07 (219.60, 303.15) | 902947.14 (774180.27, 1042155.33) | 310.04 (264.79, 358.92) | 0.87 (0.76 to 0.97) |
| North Africa and Middle East | 160950.67 (125886.71, 201566.45) | 101.54 (80.69, 125.48) | 598443.47 (476991.10, 731975.69) | 151.00 (120.98, 183.91) | 1.37 (1.31 to 1.43) |
| Oceania | 1729.53 (1263.59, 2291.78) | 55.12 (41.17, 72.00) | 4994.03 (3725.34, 6466.16) | 65.30 (49.41, 83.76) | 0.47 (0.42 to 0.52) |
| South Asia | 775864.65 (586712.35, 1009125.57) | 147.46 (112.91, 190.12) | 2242591.92 (1721658.26, 2887252.62) | 198.27 (153.13, 254.11) | 1.09 (0.95 to 1.23) |
| Southeast Asia | 152122.88 (1109069.03, 200381.82) | 65.39 (48.55, 84.96) | 417517.49 (314066.84, 537611.97) | 85.34 (64.09, 110.06) | 0.95 (0.91 to 0.99) |
| Southern Latin America | 56348.34 (43726.70, 70754.44) | 190.62 (148.16, 238.94) | 148950.24 (118601.02, 182731.68) | 311.12 (246.33, 383.38) | 1.53 (1.42 to 1.64) |
| Southern Sub-Saharan Africa | 90833.90 (73265.44, 110443.69) | 375.27 (307.05, 450.37) | 162959.16 (132507.25, 197550.67) | 332.37 (272.00, 400.72) | −0.26 (−0.33 to −0.19) |
| Tropical Latin America | 186286.12 (143578.72, 233821.14) | 224.84 (175.53, 279.41) | 365358.79 (289975.65, 449400.34) | 225.52 (178.29, 278.24) | 0.25 (0.17 to 0.32) |
| Western Europe | 743246.43 (590101.27, 925708.76) | 262.56 (206.98, 328.82) | 1057774.65 (851840.76, 1294196.21) | 299.21 (237.41, 370.94) | 0.46 (0.41 to 0.52) |
| Western Sub-Saharan Africa | 46785.96 (33485.35, 62919.32) | 58.13 (42.46, 76.96) | 156764.39 (114450.23, 206262.73) | 74.88 (55.97, 96.91) | 0.89 (0.76 to 1.03) |

*(Continued)*

Table 1. (Continued)

| Location | 1990 Number | 1990 ASR | 2021 Number | 2021 ASR | EAPC (95% CI) |
|---|---|---|---|---|---|
| **Deaths** | | | | | |
| Global | 5870.48 (5164.65, 6705.23) | 0.21 (0.19, 0.24) | 6873.72 (5830.26, 7956.85) | 0.13 (0.11, 0.15) | -1.59 (-1.73 to -1.45) |
| High SDI | 1555.89 (1506.34, 1603.57) | 0.25 (0.24, 0.25) | 1024.90 (971.29, 1083.55) | 0.10 (0.09, 0.11) | -2.92 (-3.04 to -2.80) |
| High-middle SDI | 1475.80 (1310.12, 1714.92) | 0.22 (0.19, 0.25) | 1480.60 (1273.39, 1738.85) | 0.13 (0.11, 0.15) | -1.75 (-1.96 to -1.54) |
| Middle SDI | 1945.34 (1595.60, 2341.08) | 0.24 (0.19, 0.28) | 2820.17 (2256.75, 3295.05) | 0.15 (0.12, 0.18) | -1.20 (-1.46 to -0.95) |
| Low-middle SDI | 746.55 (504.80, 1026.52) | 0.16 (0.11, 0.22) | 1304.82 (955.69, 1766.24) | 0.13 (0.09, 0.17) | -0.68 (-0.77 to -0.60) |
| Low SDI | 140.85 (77.74, 321.60) | 0.08 (0.05, 0.18) | 239.27 (138.86, 481.89) | 0.06 (0.04, 0.12) | -0.91 (-1.03 to -0.79) |
| Andean Latin America | 35.55 (28.13, 45.33) | 0.23 (0.18, 0.29) | 60.94 (44.10, 81.73) | 0.16 (0.11, 0.21) | -1.65 (-2.00 to -1.30) |
| Australasia | 34.46 (29.89, 39.78) | 0.25 (0.22, 0.29) | 28.45 (24.43, 33.26) | 0.11 (0.09, 0.12) | -2.71 (-2.99 to -2.43) |
| Caribbean | 31.29 (24.04, 42.30) | 0.18 (0.14, 0.24) | 52.70 (39.24, 74.22) | 0.16 (0.12, 0.22) | -0.54 (-0.69 to -0.38) |
| Central Asia | 4.72 (3.67, 6.60) | 0.01 (0.01, 0.02) | 53.68 (45.00, 63.39) | 0.08 (0.07, 0.10) | 5.52 (3.76 to 7.32) |
| Central Europe | 263.39 (252.43, 275.42) | 0.27 (0.26, 0.28) | 88.98 (81.38, 97.66) | 0.08 (0.07, 0.09) | -3.89 (-4.35 to -3.43) |
| Central Latin America | 322.28 (306.00, 339.86) | 0.50 (0.48, 0.53) | 747.10 (640.77, 887.25) | 0.44 (0.38, 0.53) | -0.37 (-0.56 to -0.17) |
| Central Sub-Saharan Africa | 5.71 (1.37, 62.67) | 0.03 (0.01, 0.32) | 8.01 (1.67, 94.22) | 0.02 (0.00, 0.19) | -2.08 (-2.16 to -2.01) |
| East Asia | 1928.19 (1520.78, 2500.55) | 0.29 (0.23, 0.37) | 2180.09 (1580.43, 2805.53) | 0.16 (0.12, 0.21) | -1.50 (-1.88 to -1.13) |
| Eastern Europe | 538.42 (506.16, 565.15) | 0.29 (0.27, 0.31) | 474.46 (428.31, 524.90) | 0.24 (0.22, 0.27) | -1.29 (-1.65 to -0.92) |
| Eastern Sub-Saharan Africa | 10.48 (3.06, 99.87) | 0.02 (0.01, 0.16) | 12.09 (3.07, 137.70) | 0.01 (0.00, 0.09) | -2.68 (-2.81 to -2.54) |
| High-income Asia Pacific | 387.87 (364.04, 416.04) | 0.29 (0.27, 0.31) | 137.49 (124.14, 155.55) | 0.08 (0.07, 0.09) | -5.15 (-5.60 to -4.70) |
| High-income North America | 347.19 (332.57, 361.93) | 0.19 (0.18, 0.20) | 396.48 (368.80, 440.34) | 0.12 (0.11, 0.13) | -1.60 (-1.91 to -1.30) |
| North Africa and Middle East | 129.72 (82.95, 191.30) | 0.09 (0.06, 0.13) | 219.25 (158.77, 304.89) | 0.06 (0.04, 0.08) | -1.21 (-1.35 to -1.07) |
| Oceania | 0.02 (0.01, 0.06) | 0.00 (0.00, 0.00) | 0.07 (0.02, 0.17) | 0.00 (0.00, 0.00) | 0.45 (0.20 to 0.70) |
| South Asia | 773.17 (469.57, 1123.37) | 0.18 (0.11, 0.26) | 1312.90 (900.48, 1952.50) | 0.13 (0.09, 0.19) | -1.06 (-1.13 to -0.99) |
| Southeast Asia | 131.56 (88.08, 185.93) | 0.06 (0.04, 0.09) | 232.91 (145.09, 310.38) | 0.05 (0.03, 0.06) | -1.09 (-1.22 to -0.96) |
| Southern Latin America | 82.14 (72.55, 93.08) | 0.28 (0.25, 0.32) | 97.44 (84.71, 111.46) | 0.20 (0.17, 0.22) | -0.52 (-0.93 to -0.10) |
| Southern Sub-Saharan Africa | 125.48 (90.31, 158.20) | 0.57 (0.40, 0.71) | 160.85 (125.30, 209.07) | 0.36 (0.28, 0.47) | -1.89 (-2.30 to -1.47) |
| Tropical Latin America | 103.85 (96.39, 111.92) | 0.15 (0.14, 0.16) | 236.68 (217.18, 260.16) | 0.14 (0.13, 0.16) | 0.19 (-0.10 to 0.48) |
| Western Europe | 614.95 (590.02, 639.36) | 0.20 (0.19, 0.21) | 373.13 (351.85, 395.83) | 0.09 (0.08, 0.09) | -2.35 (-2.62 to -2.07) |
| Western Sub-Saharan Africa | 0.01 (0.01, 0.10) | 0.00 (0.00, 0.00) | 0.03 (0.01, 0.18) | 0.00 (0.00, 0.00) | -1.03 (-1.11 to -0.96) |
| **DALYs** | | | | | |
| Global | 991261.27 (710641.04, 1361865.03) | 34.08 (24.59, 46.56) | 1845296.36 (1280547.77, 2593947.72) | 34.54 (23.90, 48.67) | 0.15 (0.10 to 0.20) |
| High SDI | 275776.25 (198513.96, 376107.22) | 44.71 (32.11, 61.10) | 381309.59 (265732.70, 533044.13) | 43.01 (29.66, 60.66) | 0.01 (-0.04 to 0.07) |
| High-middle SDI | 239774.61 (173059.45, 329500.75) | 35.20 (25.42, 48.34) | 404381.69 (278317.88, 569289.75) | 37.85 (25.75, 53.79) | 0.29 (0.24 to 0.34) |
| Middle SDI | 320778.36 (230100.10, 442278.86) | 35.95 (26.03, 49.12) | 667893.24 (467017.12, 937324.73) | 37.58 (26.17, 52.92) | 0.25 (0.19 to 0.30) |
| Low-middle SDI | 122525.00 (84774.93, 172654.64) | 23.15 (16.27, 32.14) | 306594.90 (209769.41, 436100.71) | 27.46 (18.95, 38.77) | 0.66 (0.59 to 0.72) |
| Low SDI | 31531.33 (20648.92, 46650.04) | 15.61 (10.41, 22.76) | 83932.30 (54473.84, 123459.19) | 17.57 (11.68, 25.35) | 0.45 (0.36 to 0.53) |
| Andean Latin America | 9726.07 (6588.71, 13922.41) | 53.48 (36.82, 75.59) | 32782.35 (21413.00, 47669.16) | 78.54 (51.54, 113.81) | 1.33 (1.25 to 1.40) |

*(Continued)*

Table 1. (Continued)

| Location | 1990 | | 2021 | | EAPC (95% CI) |
|---|---|---|---|---|---|
| | Number | ASR | Number | ASR | |
| Australasia | 6293.51 (4238.61, 9087.57) | 46.73 (31.47, 67.41) | 11715.71 (7481.66, 17274.78) | 48.71 (30.58, 72.73) | 0.24 (0.15 to 0.34) |
| Caribbean | 4676.15 (3251.55, 6603.97) | 25.20 (17.72, 35.20) | 9635.01 (6745.63, 13607.56) | 29.52 (20.58, 41.84) | 0.51 (0.46 to 0.56) |
| Central Asia | 8351.92 (5234.19, 12561.70) | 22.95 (14.46, 34.24) | 22879.18 (15109.05, 33099.37) | 35.87 (23.67, 51.88) | 1.65 (1.54 to 1.76) |
| Central Europe | 33778.21 (25020.80, 45594.73) | 37.09 (27.20, 50.50) | 33748.13 (22840.81, 48427.61) | 35.63 (23.75, 51.80) | −0.05 (−0.16 to 0.05) |
| Central Latin America | 44794.43 (33284.86, 60523.95) | 63.21 (47.61, 84.25) | 124430.05 (90529.50, 168991.48) | 73.56 (53.52, 99.91) | 0.47 (0.40 to 0.54) |
| Central Sub-Saharan Africa | 3124.62 (1817.92, 5711.80) | 14.58 (8.64, 26.84) | 9568.49 (5626.21, 15259.00) | 16.71 (9.99, 26.46) | 0.47 (0.37 to 0.57) |
| East Asia | 301262.43 (214651.06, 418630.59) | 42.54 (30.53, 58.75) | 527594.29 (365927.55, 742712.47) | 42.83 (29.33, 60.93) | 0.18 (0.09 to 0.27) |
| Eastern Europe | 61110.87 (46291.50, 80433.95) | 36.14 (27.06, 48.03) | 68370.29 (50417.14, 91817.86) | 39.79 (28.81, 54.20) | 0.17 (0.08 to 0.26) |
| Eastern Sub-Saharan Africa | 8590.80 (5203.64, 13587.08) | 12.29 (7.62, 19.16) | 23262.09 (14296.94, 35890.62) | 13.38 (8.44, 20.25) | 0.30 (0.24 to 0.37) |
| High-income Asia Pacific | 63340.34 (44916.81, 87716.91) | 48.29 (34.12, 67.15) | 63117.86 (42026.46, 91277.49) | 40.41 (26.49, 59.25) | −0.52 (−0.57 to −0.46) |
| High-income North America | 76208.19 (54636.31, 103827.99) | 41.18 (29.54, 56.11) | 132681.80 (94181.68, 181404.32) | 45.27 (31.82, 62.37) | 0.55 (0.43 to 0.66) |
| North Africa and Middle East | 27413.57 (18890.23, 38914.77) | 17.43 (12.16, 24.53) | 89425.78 (59500.26, 129010.46) | 22.66 (15.16, 32.57) | 0.95 (0.87 to 1.03) |
| Oceania | 246.33 (141.26, 392.00) | 7.81 (4.54, 12.29) | 705.53 (409.52, 1117.97) | 9.18 (5.36, 14.45) | 0.46 (0.41 to 0.50) |
| South Asia | 130355.11 (88542.89, 186386.16) | 25.54 (17.65, 35.93) | 343640.53 (230333.23, 496977.44) | 30.65 (20.71, 44.03) | 0.69 (0.57 to 0.80) |
| Southeast Asia | 26583.83 (17707.21, 38859.05) | 11.52 (7.77, 16.67) | 66379.54 (43541.04, 97341.01) | 13.57 (8.89, 19.93) | 0.58 (0.56 to 0.60) |
| Southern Latin America | 10665.96 (7604.91, 14778.95) | 36.13 (25.82, 50.00) | 23415.18 (15988.76, 33491.50) | 48.76 (33.09, 70.06) | 1.06 (0.94 to 1.19) |
| Southern Sub-Saharan Africa | 17809.40 (13158.95, 23617.25) | 73.92 (55.07, 97.30) | 27923.81 (20079.52, 37951.93) | 57.79 (41.90, 77.99) | −0.85 (−0.96 to −0.73) |
| Tropical Latin America | 29415.55 (20124.87, 41990.46) | 35.87 (24.89, 50.68) | 57977.90 (39922.05, 81277.62) | 35.73 (24.51, 50.19) | 0.24 (0.16 to 0.33) |
| Western Europe | 120971.87 (85295.77, 167346.40) | 42.33 (29.57, 58.99) | 154264.07 (104650.67, 219214.51) | 43.37 (29.01, 62.43) | 0.17 (0.13 to 0.21) |
| Western Sub-Saharan Africa | 6542.11 (3904.40, 10328.90) | 8.06 (4.89, 12.59) | 21778.77 (13147.91, 33573.67) | 10.31 (6.32, 15.71) | 0.87 (0.74 to 0.99) |

ASR, age-standardized rate; DALY, disability-adjusted life year; EAPC, estimated annual percentage change; 95% CI, 95% confidence intervals; SDI, Socio-demographic Index.

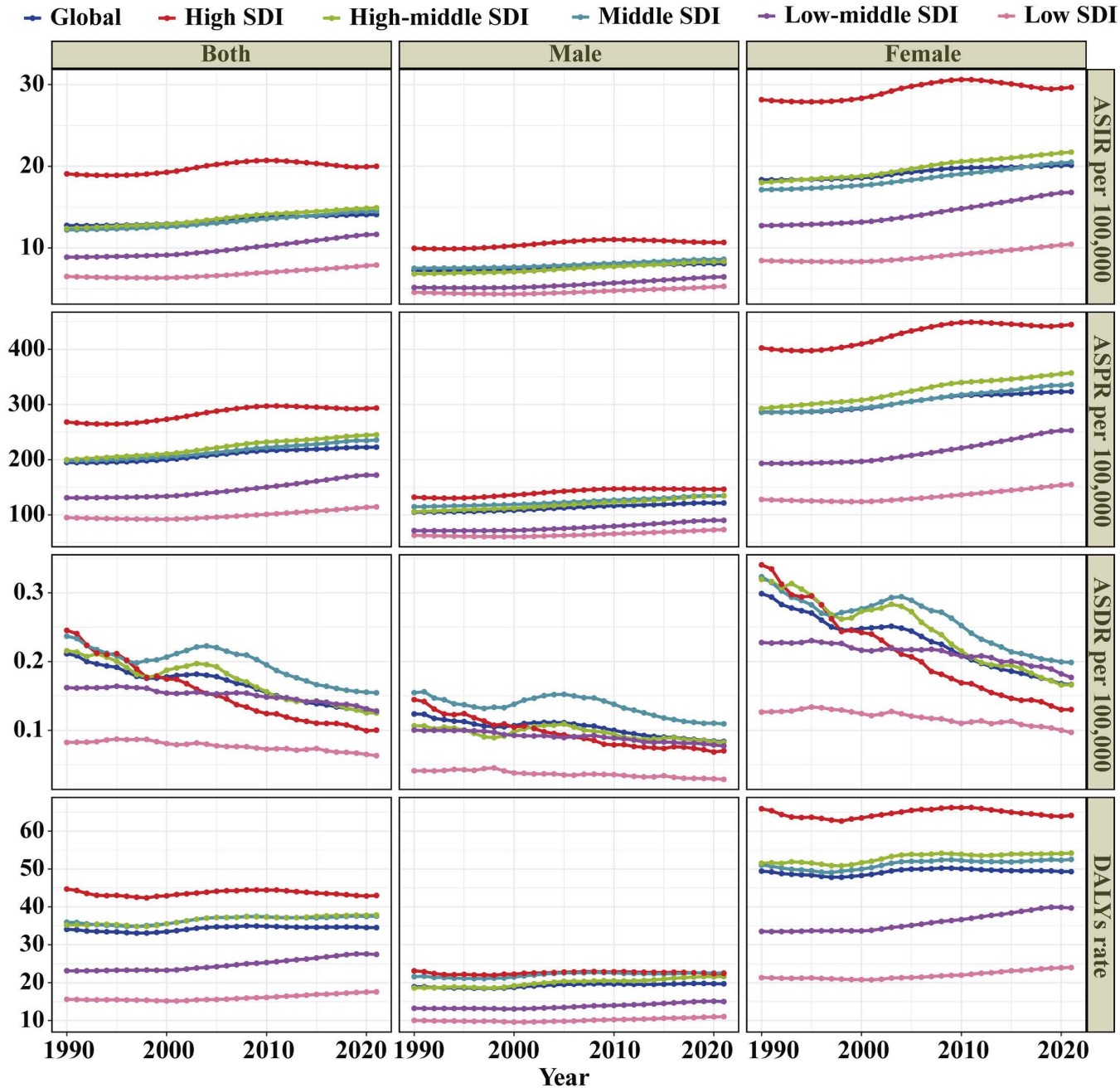

**Fig 1. Trends in rheumatoid arthritis ASIR, ASPR, ASDR and age-standardized DALYs rate among the working-age population by sex and SDI from 1990 to 2021.**

RA ASDR are notably higher in Central Latin America, Southern Sub-Saharan Africa, and Southern Latin America, with rates of 0.44 (95% UI: 0.38−0.53), 0.36 (95% UI: 0.28−0.47), and 0.20 (95% UI: 0.17−0.22), respectively. Between 1990 and 2021, the RA ASDR in Central Asia saw the fastest increase, with an EAPC of 5.52 (95% CI: 3.76 to 7.32), followed by Oceania and Tropical Latin America, while a declining trend was observed in other regions (**Table 1**). In contrast, the regions with the highest age-standardized DALYs rate for RA are Andean Latin America, with 78.54 (95% UI:

51.54−113.81); Central Latin America, at 73.56 (95% UI: 53.52−99.91); and Southern Sub-Saharan Africa, at 57.79 (95% UI: 41.90−77.99) (Table 1). From 1990 to 2021, the age-standardized DALYs rate for RA increased most significantly in Andean Latin America, with an EAPC of 1.33 (95% CI: 1.25 to 1.40), while the largest decline occurred in Southern Sub-Saharan Africa, with an EAPC of −0.85 (95% CI: −0.96 to −0.73) (Table 1).

## National burden

At the national level, the burden of RA among the working-age population is positively correlated with the SDI level, with significant differences observed between countries (S3 Fig). The ASIR of RA among the working-age population ranges from 3.18 to 41.46 per 100,000 population. Among all countries, Ireland (41.46; 95% UI: 25.74−59.78), Finland (34.37; 95% UI: 22.91−47.77), and Peru (33.78; 95% UI: 21.08−49.16) exhibit the highest ASIRs (S1 Table). In terms of the incidence cases, China (182,850.18; 95% UI: 111,605.46−269,761.76), India (136,982.18; 95% UI: 83,652.91−204,521.60), and the United States (53,161.14; 95% UI: 35,553.83−72,682.18) occupy the top three positions (S1 Table). Equatorial Guinea shows the fastest increase in ASIR (EAPC: 2.18; 95% CI: 1.99 to 2.38) (S1 Table). The distribution of the prevalent cases of RA is shown in S2 Table. The top three countries with the number of cases of RA are still China (3,213,945.13; 95% UI: 2,543,960.36−3,984,362.44), India (1,751,208.90; 95% UI: 1,332,863.55−2,276,931.59), and the United States (795,664.84; 95% UI: 670,615.59−931,103.59). Peru has the highest ASPR (681.38; 95% UI: 555.23−824.11), while Indonesia has the lowest (54.65; 95% UI: 39.78−72.33) (S1 Table). Similarly, Equatorial Guinea shows the fastest increase in ASPR (EAPC: 2.20; 95% CI: 2.01 to 2.40) (S2 Table). These findings highlight the rapid rise in the RA burden in countries such as Equatorial Guinea.

The ASDR and age-standardized DALYs rate for RA among the working-age population in 2021 show significant consistency in the most affected countries/regions. Mexico and Honduras are prominent in both indicators. Honduras leads in ASDR, with a rate of 0.62 per 100,000 population (95% UI: 0.34−1.10), followed by Mexico (0.60; 95% UI: 0.50−0.71) and Lithuania (0.60; 95% UI: 0.48−0.72). Peru ranks first in age-standardized DALYs rate with 96.74 (95% UI: 61.52−143.40), while Mexico and Honduras are ranked second and third, with 93.09 (95% UI: 68.22−125.52) and 80.80 (95% UI: 54.58−114.00), respectively (S3 and S4 Table). This overlap indicates that these countries face a high RA burden among the working-age population, affecting mortality and overall health-related quality of life in their populations. In terms of time trends, Turkmenistan (EAPC: 10.37; 95% CI: 8.60 to 12.17) and Kazakhstan (EAPC: 8.54; 95% CI: 4.31 to 12.94) exhibit the fastest increases in ASDR, while the Republic of Korea (EAPC: −5.61; 95% CI: −5.90 to −5.31) and Singapore (EAPC: −5.52; 95% CI: −6.04 to −4.99) show the largest declines (S3 Table). Kuwait demonstrates the fastest increase in age-standardized DALYs rate, with an EAPC of 2.08 (95% CI: 1.91 to 2.25) (S4 Table). Regarding the cases of DALYs, China (512,147.29; 95% UI: 353,836.57−722,685.52), India (267,578.31; 95% UI: 178,280.97−387,381.83), and the United States (116,941.67; 95% UI: 82,374.38−160,696.40) remain the top three (S4 Table). However, in terms of deaths, Mexico (521.83; 95% UI: 435.78−622.92) surpasses the United States (356.26; 95% UI: 331.89−400.91) to take third place (S3 Table).

## Sex and age patterns

From 1990 to 2021, the global age-specific incidence (S4 Fig), prevalence (S5 Fig), and DALYs rates (S6 Fig) demonstrated an upward trend across all SDI regions, with mortality rates (S7 Fig) being the sole metric showing a decline. Among the working-age population, females consistently exhibited higher rates of RA across all metrics—incidence (S4 Fig), prevalence (S5 Fig), DALYs (S6 Fig), and mortality (S7 Fig)—compared to males. These rates progressively increased with age, reaching their apex in the 60−64-year-age group. Similarly, regarding absolute case numbers, there was a year-on-year increase in incident cases (S8 Fig), prevalent cases (S9 Fig), and DALYs cases (S10 Fig) across all age groups, with females showing markedly higher numbers than males. Death cases displayed distinct age-stratified patterns in the overall population and within both sex groups (S11 Fig). A declining trend in death cases was observed across

six age groups from 15−19 to 40−44 years, followed by an increasing trend across four age groups from 45−49 to 60−64 years, with middle SDI regions playing a substantial contributory role in this pattern.

## Age-period-cohort analysis

The results of the age-period-cohort analysis for RA ASIR, ASPR, ASDR, and age-standardized DALYs rate among the working-age population are presented in S5 Table and **Fig 2**. After adjusting for period and birth cohort effects, age demonstrated a significant impact on the risk of RA across all metrics. The relative risks showed a marked upward trajectory, reaching their maximum in the 60−64-year-age group (S5 Table, **Fig 2**). When controlling for age and period effects, earlier birth cohorts exhibited higher risks for all metrics compared to later cohorts, with RRs showing a consistent decline (S5 Table, **Fig 2**). After adjusting for age and birth cohort effects, period effects demonstrated an increasing trend in risks across all metrics, with peak risks for ASIR, ASPR, ASDR, and age-standardized DALY rate observed during the 2017 period (S5 Table, **Fig 2**). Notably, while **Fig 2** presents the independent effects, S12−S15 Figs illustrate the interactions between pairs of factors. The analytical findings consistently underscore the paramount importance of age in RA, demonstrating it as a prominent factor in driving the burden of RA across ASIR, ASPR, ASDR, and age-standardized DALYs rate.

## Decomposition analysis

Over the past 32 years, RA incidence, prevalence, deaths, and DALYs have all increased, with overall changes of 90.52%, 108.39%, 17.09%, and 86.16%, respectively (S6 Table, **Fig 3**). These overall differences show a negative correlation with the SDI level. Aging only acts as a negative driver of the RA burden in low SDI regions, while epidemiological changes contribute negatively to deaths in all regions, as well as to DALYs in global and high SDI regions. Across all regions and indicators, the overall differences show a negative contribution only in deaths in high SDI regions. Furthermore, apart from

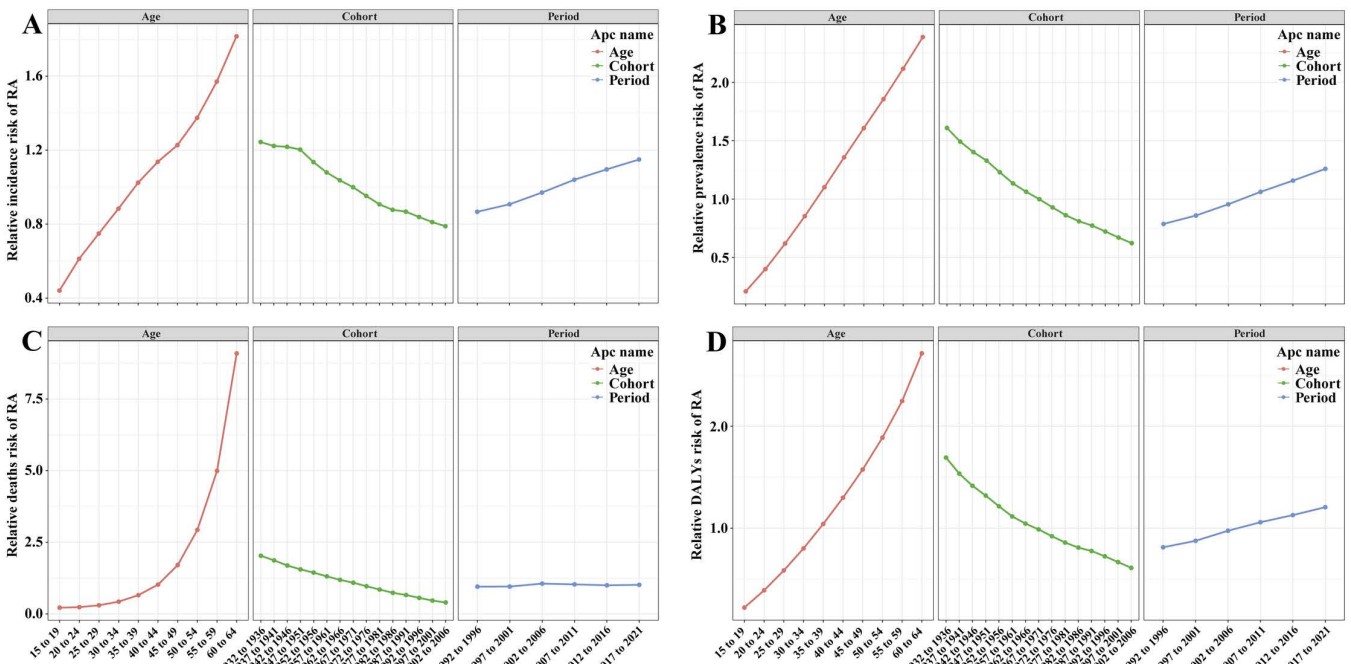

**Fig 2. The effects of age, period, and birth cohort on the relative risk of rheumatoid arthritis among the working-age population incidence (A), prevalence (B), deaths (C) and DALYs (D) rates.**

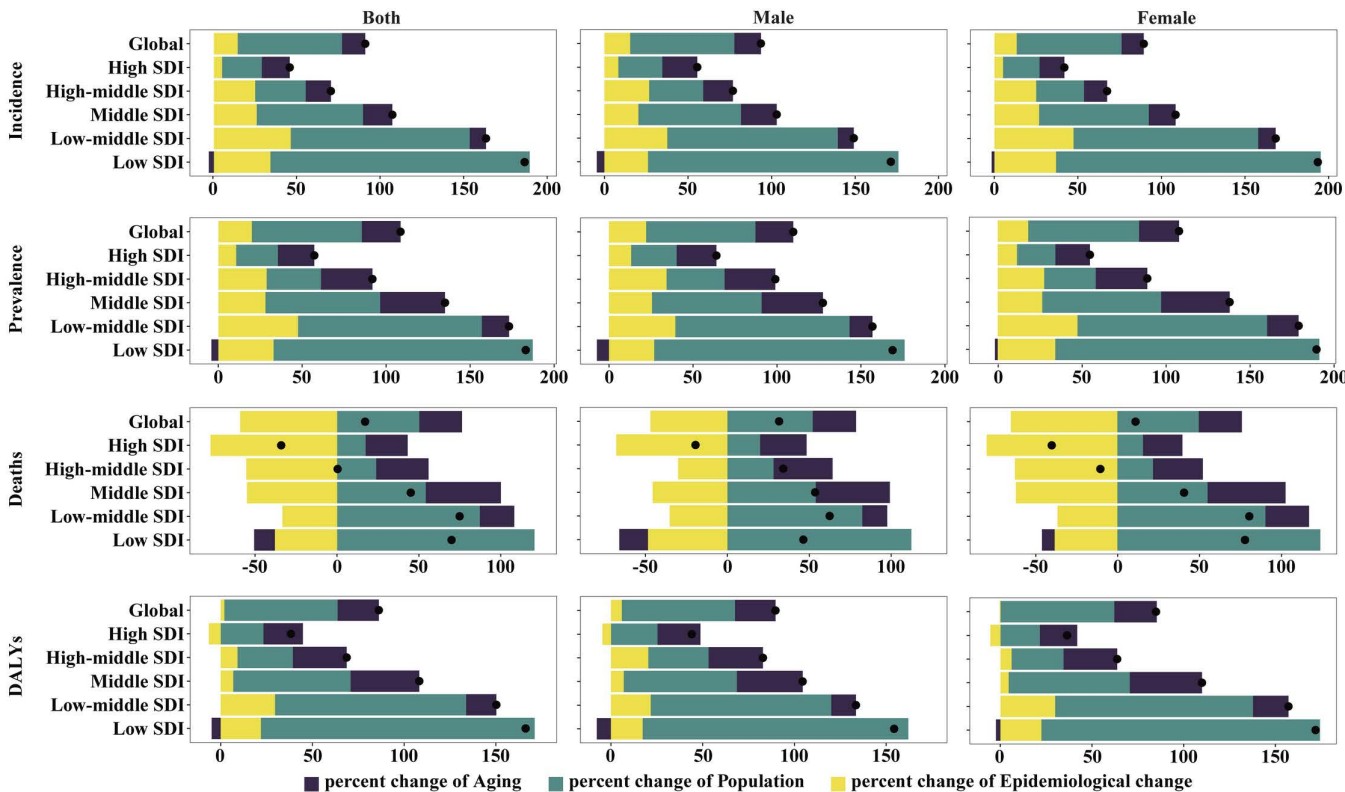

**Fig 3. Changes in incidence, prevalence, deaths and DALYs of rheumatoid arthritis among the working-age population from 1990 to 2021 by sex and SDI subgroups.** The black dots represent the overall values of changes attributable to all three components. For each component, the magnitude of positive values indicates an increase incidence, prevalence, deaths and DALYs of rheumatoid arthritis attributed to that component, while the magnitude of negative values reflects a decrease incidence, prevalence, deaths and DALYs of rheumatoid arthritis associated with that component.

deaths in certain regions, population growth has been the primary driver of RA burden in the labor force. When stratified by gender, similar trends are observed in both males and females.

## Cross-country inequality analysis

As indicated by the inequality slope index, the gap in ASIR, ASPR, and age-standardized DALY rate of RA in the labor force between high and low SDI countries has expanded from 8.31, 135.57, and 2.72 in 1990 to 10.19, 164.18, and 24.00 in 2021, respectively. This trend suggests that the burden in high SDI countries has disproportionately increased over time. However, in terms of ASDR, the gap has narrowed from 0.12 in 1990 to 0.05 in 2021, indicating a significant reduction in the death burden in high SDI countries. The concentration index, which measures relative gradient inequality, has decreased from 0.16, 0.15, 0.13, and 0.15 in 1990 to 0.14, 0.15, 0.09, and 0.14 in 2021, reflecting that while the burden remains concentrated in high SDI countries, overall health inequalities have decreased (**Fig 4**).

## Projections of global trend

It is projected that from 2022 to 2040, the global burden of RA among the working-age population will undergo significant changes, with trends differing across various indicators (**Fig 5**). The ASIR and ASPR of RA will remain relatively stable for both sexes. The ASIR is expected to slightly increase from 14.09 per 100,000 population in 2021 to 14.73 per 100,000 population in 2040, while the prevalence will increase from 222.68 to 228.19. In terms of absolute cases, the number

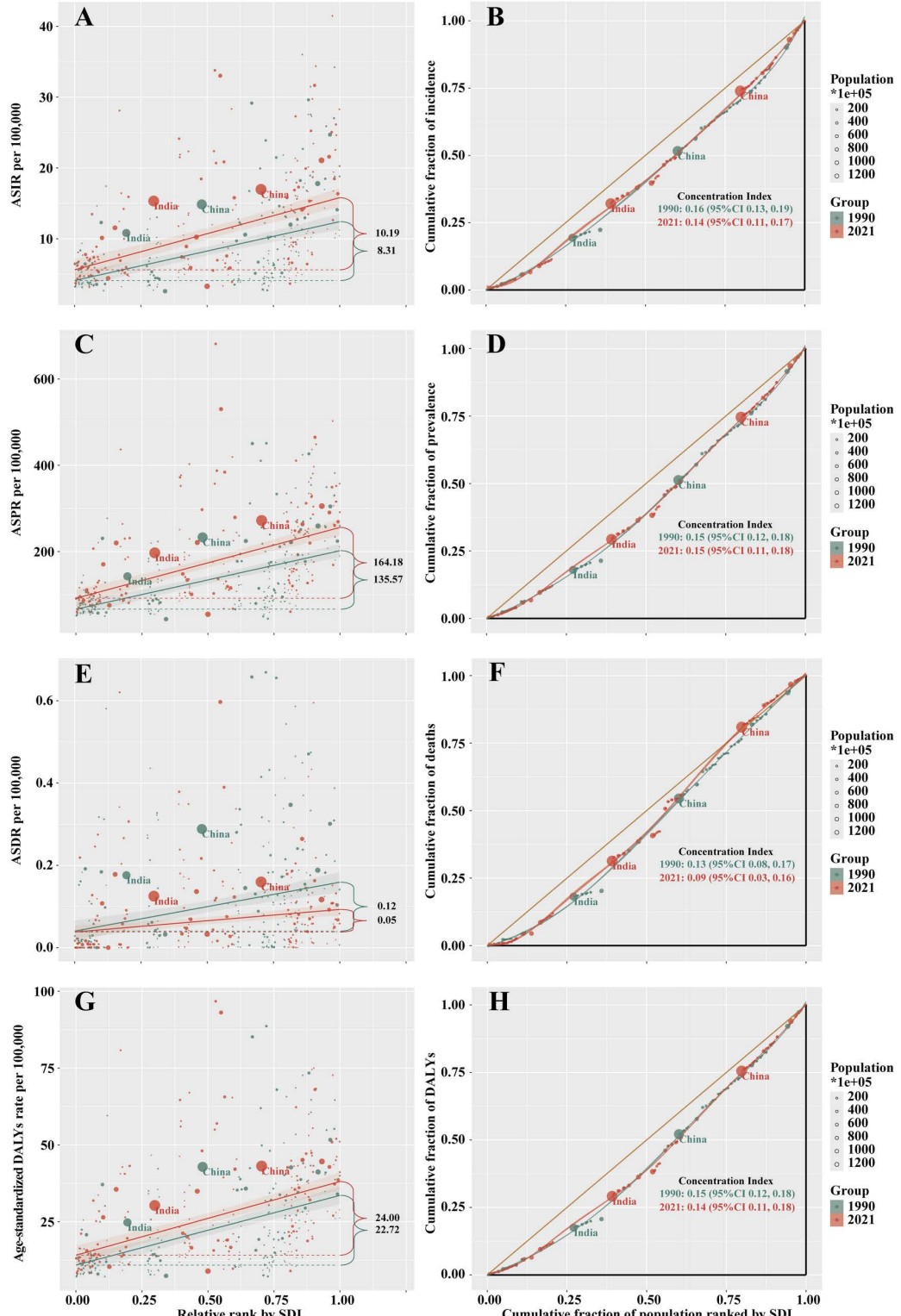

**Fig 4. The health inequality regression curves and concentration curves for rheumatoid arthritis among the working-age population in terms of ASIR (A, B), ASPR (C, D), ASDR (E, F) and age-standardized DALYs (G, H) rate from 1990 to 2021.**

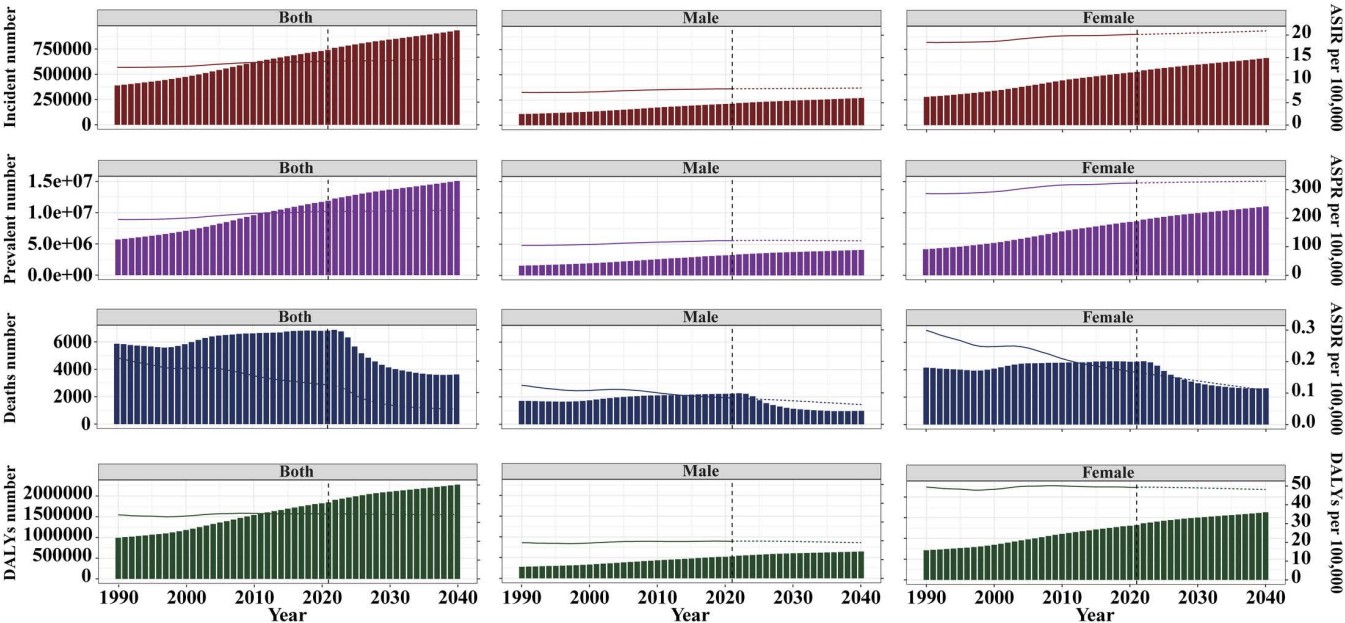

**Fig 5. Projections of the ASIR, ASPR, ASDR and age-standardized DALYs rate and cases of rheumatoid arthritis among the working-age population by sex from 2022 to 2040.**

of incident cases will rise from 740,036.40 to 933,369.60, with the number of prevalent cases increasing more notably, from 11,879,020 to 15,064,732. When stratified by gender, a slight increase in ASIR and ASPR among the working-age population is anticipated for men, but the rising trend will be more pronounced among women. The ASDR for global RA is expected to decline significantly, from approximately 0.12 per 100,000 population in 2021 to about 0.05 per 100,000 population in 2040, with the number of deaths decreasing sharply from 6,826.05 to 3,623.75. This suggests a substantial improvement in the diagnosis and treatment of RA. The age-standardized DALYs rate for global RA among the working-age population is expected to remain relatively stable for both sexes, but due to population growth, the total number of DALYs will increase from 1,845,343.50 in 2021 to 2,274,901.20 in 2040. Similar trends are also expected for both male and female populations.

## Discussion

This study offers a comprehensive assessment of the global burden of RA among the working-age population from 1990 to 2021, revealing a growing disease burden despite declining mortality rates. While age-standardized incidence and prevalence rates have shown steady increases, the decrease in mortality suggests progress in clinical management and early detection. These findings underscore the growing burden of RA and its associated disability in the global working-age population, driven by both population growth and epidemiological changes.

Regional and national disparities remain striking. Higher ASIRs, ASPRs, and DALY rates in high SDI regions may reflect both better diagnostic capacity and longer survival, whereas lower SDI regions often face underdiagnosis and limited access to care. In particular, the slower improvement in outcomes in low-income countries may stem from insufficient healthcare infrastructure, delayed treatment, and poor disease awareness. Notably, China and India carry the highest absolute burdens due to their large populations, while countries such as Ireland, Peru, and Finland report the highest incidence rates per capita. The higher RA burden observed among women and in the 60–64 age group aligns with established patterns of autoimmune susceptibility [22]. Together, these findings underscore the importance of context-specific

public health strategies. Interventions should focus on early detection, equitable healthcare access, and resource allocation tailored to regions with rising incidence or high unmet need. Addressing sociodemographic disparities will be essential for reducing the global impact of RA and improving patient outcomes.

This study demonstrates that the incidence and disease burden of RA in the global working-age population significantly increased in 2021. Compared to data from 1990, the number of new cases rose by 90.52%, and DALYs increased by 108.39%. These results are consistent with the findings of the GBD 2021 Rheumatoid Arthritis Collaborators. The study highlighted that, despite a decline in mortality rates over the past three decades, both incidence and the associated loss in quality of life have continued to rise [3]. This trend may highlight the potential for increased disease burden in the future due to population aging and lifestyle changes. Additionally, this study found significant regional disparities in the burden of RA. For example, regions with higher SDI levels have the highest ASIR, ASPR, and age-standardized DALYs rate, while middle SDI regions exhibit the lowest ASDR. This aligns with the findings of Hassen et al., who noted that while high SDI countries have made progress in reducing mortality, they face higher incidence rates and greater losses in quality of life [3,23]. Luo et al. [24] also reported a notably high burden of RA-related DALYs in high SDI regions when investigating immune-mediated inflammatory diseases. This is because high SDI regions typically have more advanced healthcare systems, better access to early diagnosis, and effective disease-modifying treatments (including biologics and targeted synthetic DMARDs). This leads to a higher reported burden, but the effective management significantly reduces mortality directly attributable to RA or its complications. And in middle SDI regions, while the overall reported burden might be lower than in high SDI regions, access to timely diagnosis, consistent monitoring, and advanced, expensive treatments may be more limited. Consequently, patients in these regions may experience more severe disease progression or complications leading to premature death, resulting in a higher ASDR [25,5].

However, countries such as Equatorial Guinea have experienced a faster growth in ASPR. This may be attributed to several factors. The acceleration of urbanization has led to shifts in lifestyle and dietary habits, potentially contributing to a rising disease burden [26,27]. Advances in healthcare infrastructure and diagnostic capabilities have facilitated improved case detection [28], while increasing life expectancy has expanded the proportion of older individuals, among whom RA prevalence is higher [29]. Additionally, environmental factors, particularly the high temperature and humidity characteristic of tropical regions, may modulate immune function and influence disease susceptibility. Genetic predisposition varies across populations, which may further account for regional disparities in RA prevalence [30].

It is noteworthy that the results indicate women are more susceptible to RA than men, a finding that has been confirmed [31,32]. This may be influenced by genetic differences, as abnormalities in the X chromosome have been linked to various autoimmune diseases and are believed to contribute to the pathogenesis of RA [33–36]. Additionally, X-linked gene mutations can result in defects in antibody production or an overactive immune system [37–39]. The disparity is not only seen in susceptibility but also in disease activity, with females experiencing more active symptoms, such as joint pain, swelling, and tissue damage, leading to more severe symptoms and higher disability [40]. Some studies suggest that the poorer condition of females may reflect gender biases in healthcare [41]. Furthermore, our study found that mortality rates among RA patients are lower in males than in females. However, a nationwide retrospective cohort study in Lithuania showed higher mortality risks in males compared to females [42], with studies on Swedish and Danish patients supporting this result [43,44]. Nonetheless, our findings align with studies conducted in countries like Germany and Spain [45,46]. There is evidence suggesting that the gender differences in mortality rates among RA patients may be due to a combination of innate and adaptive immune responses, as well as environmental, dietary, and lifestyle factors [47,48]. In terms of RA treatment, early, appropriate, and aggressive intervention is emphasized to minimize the risk of joint damage [49,50]. Indeed, men are considered an important predictor of early remission in RA, and studies suggest that they tend to respond more favorably to biologic treatments compared to females [51,52]. However, the exact reasons behind the higher remission rates in males remain unclear. It is known that estrogen helps to down-regulate inflammatory immune responses [53], and when estrogen levels rise, RA disease activity is reversed in more

than half of females [54]. Therefore, oral contraceptives and hormone replacement therapy may offer protective effects on RA disease activity [55]. Furthermore, the prevalence of RA increases with age, likely due to greater awareness of the disease among both patients and healthcare providers, as well as an increase in RA detection through routine screening.

The exact causes of RA remain uncertain; however, several risk factors have been identified. First, genetic susceptibility among relatives has been shown to increase the likelihood of developing seropositive RA [56], with patients more prone to producing autoantibodies [48]. Second, in the latest GBD study, only smoking was listed as a relevant risk factor [3]. Numerous studies have confirmed a strong correlation between smoking and RA, particularly in males and individuals with a smoking history of over 20 years [57–59]. Lastly, other modifiable risk factors have been identified in various studies, including obesity and certain dietary habits. Diets high in red meat, sugar, omega-3 fatty acids, and caffeine, as well as those low in antioxidants, have been implicated in increased RA risk [60–64]. Based on the identified risk factors for RA, relevant disease prevention consensus has been established [65–69]. The consensus emphasizes not only modifying the aforementioned risk factors but also maintaining good dental hygiene and minimizing occupational exposure to silica and dust [65]. While high SDI regions might have a higher prevalence of certain risk factors like smoking, obesity, and pro-inflammatory dietary habits, the higher reported burden of RA in these regions could also be attributed to better healthcare infrastructure, diagnostic capabilities, and surveillance systems compared to low SDI regions. In contrast, lower reported burden in low SDI regions might reflect underdiagnosis and underreporting due to limited access to healthcare, diagnostic tools, and established disease registries, rather than a genuinely lower burden of RA. Projections indicate that by 2040, the incidence, prevalence, and DALYs of RA among the working-age population will continue to rise, imposing an increasing burden on healthcare systems and societal development worldwide. Decomposition analyses suggest that this rise is primarily driven by population aging and growth. However, altering modifiable risk factors could help mitigate the growth of the RA burden to some extent.

This study has several limitations. First, the data used from the GBD database heavily relies on estimates generated by the DisMod-MR 2.1 model. Although the GBD modeling process is robust, the availability of RA-specific information is limited. Among the 204 countries included in GBD 2021, data were only available from 45 countries, and the accuracy of the estimates may be influenced by the quality and availability of data sources [3]. Second, variations in diagnostic criteria and access to diagnostic services for RA across countries and regions may introduce inconsistencies, potentially affecting the comparative analysis of RA burden. Especially the potential bias due to underreporting in lower-income countries. Third, GBD data is updated annually and may not reflect short-term fluctuations or seasonal changes in RA, potentially preventing timely development of response strategies. In addition, the EAPC assumes a constant proportional change (log-linear trend) over the entire period and can potentially mask significant fluctuations, changes in direction, or periods of acceleration/deceleration in the trend. Fourth, while the GBD database includes some RA risk factors, it may exclude others, such as dietary differences, environmental exposures, or specific medication use patterns. While the GBD study utilizes robust statistical techniques to account for these uncertainties, caution is warranted when interpreting the burden results at the national level.

## Conclusions

This study reveals significant upward trends in ASIR, ASPR, and age-standardized DALYs rate. Despite declining mortality, suggesting treatment improvements, the burden remains concentrated in high SDI regions, necessitating tailored public health strategies. Specifically, interventions should focus on strengthening early diagnosis and optimizing disease management in high SDI regions to mitigate the high burden while prioritizing enhanced access to effective treatments in middle SDI regions where mortality rates are disproportionately elevated. The greater burden observed among female and older individuals also demands targeted attention and resource allocation. With the anticipated intensification of RA's impact due to global population aging, optimizing healthcare resources and addressing modifiable risk factors are crucial.

Future research should continue to elucidate the complex interplay of genetic, environmental, and lifestyle factors contributing to RA development. Critically, addressing the observed disparities requires a concerted effort to improve healthcare infrastructure, surveillance capabilities, and equitable access to diagnosis and care, particularly in low SDI regions, to reduce inequalities and accurately assess the true global burden of RA.

## Supporting information

**S1 Fig. Trends in rheumatoid arthritis incident, prevalent, deaths and DALYs cases among the working-age population by sex and SDI from 1990 to 2021.**
(DOCX)

**S2 Fig. Associations between SDI and ASIR (A), ASPR (B), ASDR (C) and age-standardized DALYs (D) rate of rheumatoid arthritis among the working-age population across 21 GBD regions from 1990 to 2021.**
(DOCX)

**S3 Fig. Associations between SDI and ASIR (A), ASPR (B), ASDR (C) and age-standardized DALYs (D) rate of rheumatoid arthritis among the working-age population across 204 countries in 2021.**
(DOCX)

**S4 Fig. The temporal trends of rheumatoid arthritis incidence rate among the working-age population across different age groups, globally and in the SDI regions from 1990 to 2021.**
(DOCX)

**S5 Fig. The temporal trends of rheumatoid arthritis prevalence rate among the working-age population across different age groups, globally and in the SDI regions from 1990 to 2021.**
(DOCX)

**S6 Fig. The temporal trends of rheumatoid arthritis DALYs rate among the working-age population across different age groups, globally and in the SDI regions from 1990 to 2021.**
(DOCX)

**S7 Fig. The temporal trends of rheumatoid arthritis deaths rate among the working-age population across different age groups, globally and in the SDI regions from 1990 to 2021.**
(DOCX)

**S8 Fig. The temporal trends of rheumatoid arthritis incident cases among the working-age population across different age groups, globally and in the SDI regions from 1990 to 2021.**
(DOCX)

**S9 Fig. The temporal trends of rheumatoid arthritis prevalent cases among the working-age population across different age groups, globally and in the SDI regions from 1990 to 2021.**
(DOCX)

**S10 Fig. The temporal trends of rheumatoid arthritis DALYs cases among the working-age population across different age groups, globally and in the SDI regions from 1990 to 2021.**
(DOCX)

**S11 Fig. The temporal trends of rheumatoid arthritis deaths cases among the working-age population across different age groups, globally and in the SDI regions from 1990 to 2021.**
(DOCX)

**S12 Fig. The role of interaction between two factors including age-period (A), age-cohort (B), and cohort-period (C) on the incidence of rheumatoid arthritis among the working-age population.**
(DOCX)

**S13 Fig. The role of interaction between two factors including age-period (A), age-cohort (B), and cohort-period (C) on the prevalence of rheumatoid arthritis among the working-age population.**
(DOCX)

**S14 Fig. The role of interaction between two factors including age-period (A), age-cohort (B), and cohort-period (C) on the DALYs of rheumatoid arthritis among the working-age population.**
(DOCX)

**S15 Fig. The role of interaction between two factors including age-period (A), age-cohort (B), and cohort-period (C) on the deaths of rheumatoid arthritis among the working-age population.**
(DOCX)

**S1 Table. National trends in the burden of rheumatoid arthritis incidence among working-age population: 1990−2021.**
(DOCX)

**S2 Table. National trends in the burden of rheumatoid arthritis prevalence among working-age population 1990−2021.**
(DOCX)

**S3 Table. National trends in the burden of rheumatoid arthritis deaths among working-age population 1990−2021.**
(DOCX)

**S4 Table. National trends in the burden of rheumatoid arthritis DALYs among working-age population 1990−2021.**
(DOCX)

**S5 Table. Relative risks of rheumatoid arthritis incidence, prevalence, deaths and DALYs for both sexes due to age, period, and birth cohort effects.**
(DOCX)

**S6 Table. Changes in incidence, prevalence, deaths and DALYs number of rheumatoid arthritis according to population-level determinants from 1990 to 2021.**
(DOCX)

## Acknowledgments

We sincerely appreciate the IHME institution for providing the GBD data. We also thanks to JD_GBDR (V2.34.2, Jingding Medical Technology Co., Ltd.) for his help in our exploration of GBD database.

## Author contributions

**Conceptualization:** Jun Li.

**Data curation:** Jun Li, Zhiyong Li, Chengluo Hao.

**Formal analysis:** Jun Li.

**Methodology:** Jun Li.

**Software:** Jun Li, Zhiyong Li.

**Supervision:** Chengluo Hao, Xiangrui Chen.

**Visualization:** Jun Li, Zhiyong Li.

**Writing – original draft:** Jun Li.

**Writing – review & editing:** Jun Li.

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
