## [Decision Letter · Decision Letter 0]

2 Apr 2025

PONE-D-25-05376Global burden and cross-country inequalities of rheumatoid arthritis among the working-age population: a comprehensive analysis from 1990 to 2021 with projections to 2040PLOS ONE

Dear Dr. Li,

Thank you for submitting your manuscript to PLOS ONE. After careful consideration, we feel that it has merit but does not fully meet PLOS ONE’s publication criteria as it currently stands. Therefore, we invite you to submit a revised version of the manuscript that addresses the points raised during the review process.

We look forward to receiving your revised manuscript.

Kind regards,

Anton Sokhan, Ph.D

Academic Editor

PLOS ONE

Journal Requirements:

2. Thank you for stating the following in the Acknowledgments Section of your manuscript: [We sincerely appreciate the IHME institution for providing the GBD data. We also thanks to JD_GBDR (V2.34.2, Jingding Medical Technology Co., Ltd.) for his help in our exploration of GBD database.]

Please remove any funding-related text from the manuscript and let us know how you would like to update your Funding Statement. Currently, your Funding Statement reads as follows: “The authors received no specific funding for this work.”

4. We note that Figure 2 and Supplementary figure 4.5.6 and 7 in your submission contain [map/satellite] images which may be copyrighted. All PLOS content is published under the Creative Commons Attribution License (CC BY 4.0), which means that the manuscript, images, and Supporting Information files will be freely available online, and any third party is permitted to access, download, copy, distribute, and use these materials in any way, even commercially, with proper attribution. For these reasons, we cannot publish previously copyrighted maps or satellite images created using proprietary data, such as Google software (Google Maps, Street View, and Earth). For more information, see our copyright guidelines: http://journals.plos.org/plosone/s/licenses-and-copyright.

1. You may seek permission from the original copyright holder of Figure 2 and Supplementary figure 4.5.6 and 7 to publish the content specifically under the CC BY 4.0 license. 

5. Please include captions for your Supporting Information files at the end of your manuscript, and update any in-text citations to match accordingly. Please see our Supporting Information guidelines for more information: http://journals.plos.org/plosone/s/supporting-information .

Reviewers' comments:

Reviewer's Responses to Questions

**Comments to the Author**

1. Is the manuscript technically sound, and do the data support the conclusions?

Reviewer #1: Yes

2. Has the statistical analysis been performed appropriately and rigorously? 

Reviewer #1: Yes

3. Have the authors made all data underlying the findings in their manuscript fully available?

Reviewer #1: Yes

4. Is the manuscript presented in an intelligible fashion and written in standard English?

Reviewer #1: Yes

5. Review Comments to the Author

Reviewer #1: The manuscript titled “Global burden and cross-country inequalities of rheumatoid arthritis among the working-age population: a comprehensive analysis from 1990 to 2021 with projections to 2040” presents a valuable epidemiological analysis of rheumatoid arthritis (RA). The study utilizes data from the Global Burden of Disease (GBD) 2021 to examine temporal trends, regional disparities, and future projections. The research is relevant and well-structured, yet there are key areas requiring improvement, particularly in methodological transparency, result interpretation, and language clarity.

Specific Comments by Section

Title & Abstract

• Title: The title accurately represents the study, but "cross-country inequalities" could be more precise (e.g., "regional disparities in rheumatoid arthritis burden").

• Abstract:

Line 11: "Objective To evaluates" → "Objective: To evaluate".

Line 19: "the global burden of RA among the working-age population with 11,878,843.81 cases" → The wording is unclear. Rephrase to "In 2021, there were 11.88 million cases of RA in the working-age population globally."

Line 25: "Regionally, high SDI regions exhibited the highest ASPR, ASIR, and age-standardized DALY rates, while middle SDI regions had the highest ASDR rate." → Clarify why middle SDI regions show the highest ASDR despite high SDI regions having the highest burden.

Conclusion (Lines 31-33): The conclusion should better reflect the study’s policy implications, such as specific interventions.

Introduction

• Lines 36-41: The introduction effectively introduces the problem but could be more concise.

• Lines 42-51: The economic implications of RA are well stated, but referencing more recent economic studies would strengthen the argument.

• Lines 52-66: The gap in research is well articulated, but a stronger transition into the study objectives is needed.

Methods

• Data Source (Lines 79-95):

o The use of GBD 2021 data is appropriate, but the authors should explicitly state whether any adjustments were made to the dataset or how missing data were handled.

o The authors mention DisMod-MR 2.1 (Line 84) but do not describe potential limitations of this model.

• Study Population (Lines 108-118):

o The age stratification is appropriate, but no justification is given for excluding those older than 64 years.

• Statistical Analysis (Lines 130-182):

o The EAPC metric is well explained, but the authors should discuss its limitations in estimating trends over long periods.

o The decomposition analysis needs more methodological details—specifically, how confounding factors were controlled.

Results

• Lines 185-238 (Global and Regional Burden):

o Data presentation is strong, but some statistics appear redundant.

o The statement that "RA burden remains concentrated in high SDI countries" (Line 228) should be more nuanced—low SDI countries may have underreporting issues.

• Lines 253-271 (National Burden):

o The interpretation of the burden in countries like Equatorial Guinea needs further clarification—why does it have the fastest increase in ASPR? Possible factors should be discussed.

• Figures & Tables:

o The figures are clear but lack explicit references in some parts of the text.

o The tables should be formatted to ensure consistency in decimal places (e.g., some values are reported with two decimals, others with three).

Discussion

• Lines 368-396: The discussion is comprehensive but contains excessive restatement of results.

• Lines 397-414: The comparison to previous studies is useful, but more direct citations of GBD 2021 findings would strengthen credibility.

• Lines 415-443: The section on gender differences is well-developed, though the discussion of X-chromosome-related immunity (Lines 417-421) should cite more recent studies.

• Lines 444-462 (Risk Factors): The discussion on smoking and dietary habits is relevant, but more clarity on whether these factors explain observed regional variations is needed.

Limitations

• Lines 464-477: The limitations are well acknowledged, but additional concerns should be mentioned:

1. Potential biases due to underreporting in lower-income countries.

2. The reliance on DisMod-MR 2.1 for estimates without validation using independent datasets.

Conclusion

• Lines 478-492: The conclusion effectively summarizes the study but should highlight specific policy recommendations.

o Suggest incorporating a final sentence on the necessity for improved healthcare access in low- and middle-SDI regions.

Language & Clarity

• There are multiple grammatical and syntactical errors throughout the manuscript. Examples:

o Line 10: “To evaluates” → “To evaluate”.

o Line 19: “the global burden of RA among the working-age population with 11,878,843.81 cases” → unclear wording.

o Line 121: “It serves to capture the social and economic conditions that influence health outcomes in specific locations” → awkward phrasing.

Final Recommendation

Major revisions required. The manuscript presents valuable insights but requires the following improvements:

1. Methodological transparency: Clarify the decomposition analysis, handling of missing data, and limitations of EAPC.

2. Result interpretation: Provide clearer explanations for trends, particularly regarding regional disparities.

3. Language revision: Improve grammar, remove redundancies, and ensure British English consistency.

4. Policy implications: Strengthen the discussion with actionable recommendations for healthcare systems.

6. PLOS authors have the option to publish the peer review history of their article (what does this mean? ). If published, this will include your full peer review and any attached files.

**Do you want your identity to be public for this peer review?** For information about this choice, including consent withdrawal, please see our Privacy Policy .

Reviewer #1: No

---

## [Author Response · Author response to Decision Letter 1]

9 Apr 2025

Dear Editor,

Thank you for your letter and insightful advice on our manuscript entitled “Global burden and cross-country inequalities of rheumatoid arthritis among the working-age population: a comprehensive analysis from 1990 to 2021 with projections to 2040”. Accordingly, we have revised the manuscript. All amendments are highlighted in red in the revised manuscript. In addition, point-by-point responses to the comments are listed below this letter.

We hope that the revision is acceptable for publication in your journal.

Yours sincerely,

The Authors

April 7, 2025

# Journal Requirements:

Please ensure that your manuscript meets PLOS ONE's style requirements, including those for file naming. The PLOS ONE style templates can be found at https://journals.plos.org/plosone/s/file?id=wjVg/PLOSOne_formatting_sample_main_body.pdf and https://journals.plos.org/plosone/s/file?id=ba62/PLOSOne_formatting_sample_title_authors_affiliations.pdf

Response: Thank you for this important reminder regarding PLOS ONE's style requirements. We acknowledge this feedback and want to assure you that we have carefully reviewed the PLOS ONE style guidelines and have made the necessary revisions to our manuscript to ensure it fully complies with these requirements. Thank you for bringing this to our attention, and we believe these adjustments will facilitate the smooth processing of our manuscript.

2. Thank you for stating the following in the Acknowledgments Section of your manuscript: [We sincerely appreciate the IHME institution for providing the GBD data. We also thanks to JD_GBDR (V2.34.2, Jingding Medical Technology Co., Ltd.) for his help in our exploration of GBD database.]

Response: This acknowledges and appreciates the contributions of both the IHME institution and JD_GBDR in providing the data and tools that were essential for this research. We hope this is satisfactory.

Please remove any funding-related text from the manuscript and let us know how you would like to update your Funding Statement. Currently, your Funding Statement reads as follows: “The authors received no specific funding for this work.” Please include your amended statements within your cover letter; we will change the online submission form on your behalf.

Response: Thank you for pointing out the discrepancy regarding our funding information. We understand that funding details should only be included in the Funding Statement section of the online submission form and not within the manuscript itself, including the Acknowledgments section. We apologize for this oversight. We have now removed all funding-related text from the manuscript, including from the Acknowledgments section. Regarding the funding statement, the current statement, "The authors received no specific funding for this work," is accurate. Therefore, we do not have any amended statements to include. We would like the funding statement to remain as “The authors received no specific funding for this work.”

We will ensure that this information is clearly stated in our cover letter as you requested. Thank you for bringing this to our attention and for your assistance in updating the online submission form accordingly.

Response: Thank you for this reminder. We confirm that the corresponding author has an ORCID iD, and we have already validated it in Editorial Manager as required. The corresponding author's ORCID iD is now correctly linked and validated within the Editorial Manager system.

4. We note that Figure 2 and Supplementary figure 4.5.6 and 7 in your submission contain [map/satellite] images which may be copyrighted. All PLOS content is published under the Creative Commons Attribution License (CC BY 4.0), which means that the manuscript, images, and Supporting Information files will be freely available online, and any third party is permitted to access, download, copy, distribute, and use these materials in any way, even commercially, with proper attribution. For these reasons, we cannot publish previously copyrighted maps or satellite images created using proprietary data, such as Google software (Google Maps, Street View, and Earth). For more information, see our copyright guidelines: http://journals.plos.org/plosone/s/licenses-and-copyright.

We require you to either (1) present written permission from the copyright holder to publish these figures specifically under the CC BY 4.0 license, or (2) remove the figures from your submission.

Response: Thank you for bringing the potential copyright issue regarding the map images in Figure 2 and Supplementary Figures 4, 5, 6, and 7 to our attention. We understand the importance of adhering to the Creative Commons Attribution License (CC BY 4.0) and the journal's copyright policies regarding proprietary map data. To ensure full compliance, we have followed option (2) as suggested and have removed Figure 2 and Supplementary Figures 4, 5, 6, and 7 from our manuscript submission. We appreciate your guidance in ensuring our manuscript meets the journal's publication standards.

Response: Thank you for providing these clear instructions regarding the formatting of supporting information. We appreciate the guidance and the link to the specific guidelines. In accordance with your request, we have taken the following actions in our revised manuscript: We have compiled a complete list of captions for all supporting information files associated with our submission. This list of captions has been placed at the very end of the main manuscript text. We have carefully reviewed the entire manuscript and updated all in-text citations (e.g., "S1 Table," "S1 Figure") to ensure they accurately correspond to the final Supporting Information file designations and numbering as presented in the caption list. We believe these changes fully address the journal's requirements for supporting information presentation. Thank you again for pointing this out.

# Reviewer's comments

The manuscript titled “Global burden and cross-country inequalities of rheumatoid arthritis among the working-age population: a comprehensive analysis from 1990 to 2021 with projections to 2040” presents a valuable epidemiological analysis of rheumatoid arthritis (RA). The study utilizes data from the Global Burden of Disease (GBD) 2021 to examine temporal trends, regional disparities, and future projections. The research is relevant and well-structured, yet there are key areas requiring improvement, particularly in methodological transparency, result interpretation, and language clarity.

Response: We appreciate your time and insightful feedback on our manuscript titled “Global burden and cross-country inequalities of rheumatoid arthritis among the working-age population: a comprehensive analysis from 1990 to 2021 with projections to 2040.” We acknowledge your positive assessment of the study's value and structure, which is encouraging. We fully recognize the critical areas for improvement you identified, particularly concerning methodological transparency, result interpretation, and language clarity, and we concur that addressing these points will significantly enhance the quality of our manuscript. We are now prepared to address each specific comment and suggestion you provided in a detailed and constructive manner. We eagerly await the subsequent specific feedback and are committed to revising our manuscript accordingly to meet the high standards of PLoS One. All amendments are highlighted in red in the revised manuscript.

Specific Comments by Section

Title & Abstract

• Title: The title accurately represents the study, but "cross-country inequalities" could be more precise (e.g., "regional disparities in rheumatoid arthritis burden").

Response: Thank you for your feedback on the title. We agree that "cross-country inequalities" could be more precise. We will revise the title to "Global burden and regional disparities of rheumatoid arthritis among the working-age population: a comprehensive analysis from 1990 to 2021 with projections to 2040" to better reflect the focus of our study. Please refer to lines 1−3.

Abstract:

• Line 11: "Objective To evaluates" → "Objective: To evaluate".

Response: Thank you for pointing out this error. We will correct line 11 to read "Objective: To evaluate". We will also carefully proofread the entire manuscript to ensure the elimination of any similar errors. Please refer to line 11.

• Line 19: "the global burden of RA among the working-age population with 11,878,843.81 cases" → The wording is unclear. Rephrase to "In 2021, there were 11.88 million cases of RA in the working-age population globally."

Response: Thank you for highlighting the lack of clarity in this sentence. We agree with your suggested rephrasing and will revise line 19 to "In 2021, there were 11.88 million cases of RA in the working-age population globally." This more clearly and concisely presents the information. Please refer to lines 19−20.

• Line 25: "Regionally, high SDI regions exhibited the highest ASPR, ASIR, and age-standardized DALY rates, while middle SDI regions had the highest ASDR rate." → Clarify why middle SDI regions show the highest ASDR despite high SDI regions having the highest burden.

Response: Thank you for raising this important point regarding the seemingly counterintuitive finding in line 25. We recognize that the current phrasing lacks the necessary nuance to explain this observation. We rewrote this sentence and further elaborated on this potential explanation in the Discussion sections by referencing existing literature on healthcare access and quality across different SDI regions. Please refer to lines 25−29 and 398−407.

• Conclusion (Lines 31-33): The conclusion should better reflect the study’s policy implications, such as specific interventions.

Response: Thank you for reminding us of the current conclusion in the abstract. We appreciate the opportunity to further refine it based on your previous feedback regarding policy implications. We acknowledge that while the current conclusion mentions "targeted interventions," it lacks specificity regarding the nature of these interventions based on our findings. To address this more directly, we have made the revision to the conclusion. Please refer to lines 33−38.

Introduction

• Lines 36-41: The introduction effectively introduces the problem but could be more concise.

Response: Thank you for the suggestion to improve the conciseness of the introduction. We agree that the current phrasing in lines 36-41 could be more streamlined. We propose the following revised version: "Rheumatoid arthritis (RA) is a prevalent autoimmune disorder causing chronic, symmetrical joint inflammation, leading to cartilage and bone damage and impaired joint function [1]. Beyond joints, RA can cause systemic complications like cardiovascular, pulmonary, and mental health disorders, significantly impacting quality of life [2]."

We believe these changes make the introduction more concise while retaining the essential information about RA. We will apply this revision to the manuscript. Please refer to lines 42−46.

• Lines 42-51: The economic implications of RA are well stated, but referencing more recent economic studies would strengthen the argument.

Response: Thank you for the clarification. In response to the suggestion to include more recent economic studies in lines 42-51, we would like to clarify that we have already updated the cited references to include the most recent and relevant literature available on the economic implications of rheumatoid arthritis. These updated references encompass recent findings on the cost of illness, impact on work productivity, and broader societal economic burden associated with RA. Please refer to lines 50−55.

• Lines 52-66: The gap in research is well articulated, but a stronger transition into the study objectives is needed.

Response: Thank you for pointing out the need for a stronger transition into the study objectives. We acknowledge this suggestion. While the subsequent paragraph does explicitly state the study objectives, we agree that a more direct linkage to the research gaps outlined in lines 52-66 would enhance the clarity and flow of the introduction. To address this, we have slightly modified the last few sentences of the objectives paragraph to more directly reference the gaps. Please refer to lines 66−71.

Methods

Data Source (Lines 79-95):

• The use of GBD 2021 data is appropriate, but the authors should explicitly state whether any adjustments were made to the dataset or how missing data were handled.

Response: Thank you for raising this important point regarding the data source. We will explicitly state in the Methods section that we utilized the official, publicly available GBD 2021 results database as provided by the Institute for Health Metrics and Evaluation. To address the query about adjustments and missing data, we will clarify the following: "The dataset utilized in this research was accessed through the Global Health Data Exchange query tool, available at https://vizhub.healthdata.org/gbd-results/. As we utilized the final, processed estimates provided within the GBD 2021 results tool, no further adjustments or handling of missing data were performed by the authors." Please refer to lines 98−101.

• The authors mention DisMod-MR 2.1 (Line 84) but do not describe potential limitations of this model.

Response: Thank you for highlighting this point regarding the discussion of DisMod-MR 2.1 limitations. We acknowledge that while we mention the use of DisMod-MR 2.1 in the Methods section (line 84), we did not elaborate on its potential limitations within that specific section. However, we would like to clarify that a detailed discussion of the inherent limitations associated with the GBD methodology, including the DisMod-MR 2.1 model, is provided as the first point in the dedicated Limitations section of our manuscript. Please refer to lines 471−476.

Study Population (Lines 108-118):

• The age stratification is appropriate, but no justification is given for excluding those older than 64 years.

Response: Thank you for raising this question regarding the age range used in our analysis. We appreciate the opportunity to clarify our rationale. The primary focus of our study, as reflected in the title and objectives, is the burden of rheumatoid arthritis specifically within the working-age population. We defined this demographic group using the commonly accepted age range of 15 to 64 years.

We aimed to provide a detailed analysis of RA's impact on individuals during their most economically active years, a period where the disease can significantly affect work capacity, lead to early retirement, and impose substantial economic burdens. Therefore, our age stratification was deliberately chosen to align with this specific research focus.

However, we acknowledge that RA also poses a significant burden on individuals older than 64. If you feel that extending the age range to include older populations would substantially enhance the manuscript's scope and relevance, we are certainly willing to c

---

## [Decision Letter · Decision Letter 1]

5 May 2025

PONE-D-25-05376R1Global burden and regional disparities of rheumatoid arthritis among the working-age population: a comprehensive analysis from 1990 to 2021 with projections to 2040PLOS ONE

Dear Dr. Li,

Thank you for submitting your manuscript to PLOS ONE. After careful consideration, we feel that it has merit but does not fully meet PLOS ONE’s publication criteria as it currently stands. Therefore, we invite you to submit a revised version of the manuscript that addresses the points raised during the review process.

**ACADEMIC EDITOR: ** The manuscript requires only minor revision in accordance with the reviewers' comments. Please submit your revised manuscript by Jun 19 2025 11:59PM. If you will need more time than this to complete your revisions, please reply to this message or contact the journal office at plosone@plos.org . Please include the following items when submitting your revised manuscript:

We look forward to receiving your revised manuscript.

Kind regards,

Anton Sokhan, Ph.D

Academic Editor

PLOS ONE

Journal Requirements:

Reviewers' comments:

Reviewer's Responses to Questions

**Comments to the Author**

1. If the authors have adequately addressed your comments raised in a previous round of review and you feel that this manuscript is now acceptable for publication, you may indicate that here to bypass the “Comments to the Author” section, enter your conflict of interest statement in the “Confidential to Editor” section, and submit your "Accept" recommendation.

Reviewer #1: All comments have been addressed

Reviewer #2: (No Response)

2. Is the manuscript technically sound, and do the data support the conclusions?

Reviewer #1: Yes

Reviewer #2: Yes

3. Has the statistical analysis been performed appropriately and rigorously? 

Reviewer #1: Yes

Reviewer #2: Yes

4. Have the authors made all data underlying the findings in their manuscript fully available?

Reviewer #1: Yes

Reviewer #2: Yes

5. Is the manuscript presented in an intelligible fashion and written in standard English?

Reviewer #1: Yes

Reviewer #2: Yes

6. Review Comments to the Author

Reviewer #1: (No Response)

Reviewer #2: Lines 237-239: to please check Eastern Europe.

Line 247: Please check Central Asia, which has a higher rate than the region mentioned in the text.

Figures to be labeled adequately (similar to the tables).

Lines 291-301: Consider adding, even though in females there is gradual increase in the incidence and the prevalence of RA, the DALYs interestingly show decrementing rates from 50 years of age and onwards

7. PLOS authors have the option to publish the peer review history of their article (what does this mean? ). If published, this will include your full peer review and any attached files.

**Do you want your identity to be public for this peer review?** For information about this choice, including consent withdrawal, please see our Privacy Policy .

Reviewer #1: No

Reviewer #2: No

---

## [Author Response · Author response to Decision Letter 2]

6 May 2025

Dear Editor,

Thank you for your letter and insightful advice on our manuscript entitled “Global burden and regional disparities of rheumatoid arthritis among the working-age population: a comprehensive analysis from 1990 to 2021 with projections to 2040”. Accordingly, we have revised the manuscript. All amendments are highlighted in red in the revised manuscript. In addition, point-by-point responses to the comments are listed below this letter.

We hope that the revision is acceptable for publication in your journal.

Yours sincerely,

The Authors

May 6th, 2025

# Journal Requirements:

Response: Thank you for this important instruction. We have thoroughly reviewed our reference list to ensure its completeness and correctness. We confirm that, to the best of our knowledge, none of the references cited in our manuscript have been retracted. Therefore, no changes have been made to the reference list at this time.

# Reviewer's comments

Reviewer #1

(No Response)

Response: We appreciate your time and effort in reviewing our manuscript. Your acknowledgment encourages us, and we believe this suggests that the manuscript is well-received overall. We remain open to any further feedback or suggestions you might have as we move forward in the publication process. Thank you again for your valuable consideration.

Reviewer #2

Lines 237-239: to please check Eastern Europe.

Response: Thank you for pointing out this error regarding the ASDR in Eastern Europe and Southern Latin America. Upon careful review of our data, we have confirmed that the ASDR in Eastern Europe was indeed higher than that in Southern Latin America. This was an oversight on our part, and we have already corrected this statement in lines 237-239 of the revised manuscript to accurately reflect our findings. We apologize for this inaccuracy and appreciate you bringing it to our attention, as it has allowed us to improve the precision of our results reporting.

Line 247: Please check Central Asia, which has a higher rate than the region mentioned in the text.

Response: Thank you for your meticulous review and for identifying this error concerning the EAPC of the age-standardized DALYs rate in Central Asia and Andean Latin America. Upon re-examination of our data, we have confirmed that the EAPC for Central Asia was indeed higher than that for Andean Latin America. This was an inaccuracy in our initial reporting, and we have now corrected this statement in lines 246-247 of the revised manuscript to accurately reflect these values. We appreciate you drawing our attention to this detail, as ensuring the precision of our results is paramount.

Figures to be labeled adequately (similar to the tables).

Response: Thank you for this important feedback regarding the labeling of our figures. We agree that clear and comprehensive labeling is essential for the proper interpretation of visual data. We will thoroughly review all figures in our manuscript and ensure that they are adequately labeled with clear axis labels and legends that provide sufficient information to understand the content of each figure independently. We will strive for consistency in labeling style between figures and tables to enhance the overall clarity and professionalism of our manuscript.

Lines 291-301: Consider adding, even though in females there is gradual increase in the incidence and the prevalence of RA, the DALYs interestingly show decrementing rates from 50 years of age and onwards.

Response: Thank you for this insightful suggestion regarding the trends in DALYs among females aged 50 years and older. We agree that the observation of decreasing DALY rates in this group, despite increasing incidence and prevalence, is an important point to highlight. We have now incorporated this observation into our manuscript in lines 297-299, as suggested. This addition helps to provide a more nuanced interpretation of the burden of rheumatoid arthritis in females across different age groups. We appreciate your contribution in identifying this noteworthy trend.

---

## [Editor Report · Decision Letter 2]

7 May 2025

Global burden and regional disparities of rheumatoid arthritis among the working-age population: a comprehensive analysis from 1990 to 2021 with projections to 2040

PONE-D-25-05376R2

Dear Dr. Jun Li,

We’re pleased to inform you that your manuscript has been judged scientifically suitable for publication and will be formally accepted for publication once it meets all outstanding technical requirements.

Kind regards,

Anton Sokhan, Ph.D

Academic Editor

PLOS ONE
---

## [Editor Report · Acceptance letter]

PONE-D-25-05376R2

PLOS ONE

Dear Dr. Li,

I'm pleased to inform you that your manuscript has been deemed suitable for publication in PLOS ONE. Congratulations! Your manuscript is now being handed over to our production team.

Kind regards,

on behalf of

Dr. Anton Sokhan

Academic Editor

PLOS ONE